

# Implementation of an Immersed Boundary Method in the Meso-NH model: Applications to an idealized urban-like environment.

Franck Auguste[1], Géraldine Réa[1], Roberto Paoli[3,4], Christine Lac[2], Valery Masson[2], and
Daniel Cariolle[1,2]

[1]CECI, CNRS, CERFACS, Toulouse, France
[2]CNRM, CNRS, Météo-France, Toulouse, France
[3]University of Illinois at Chicago, Department of Mechanical and Industrial Engineering, Chicago, USA
[4]Argonne National Laboratory, Argonne, USA

*Correspondence to:* Franck Auguste (franck.auguste@cerfacs.fr)

**Abstract.**

   This study describes the numerical implementation, verification and validation of an Immersed Boundary Method (IBM) in the atmospheric solver Meso-NH for applications to urban flow modelling. The IBM represents the fluid-solid interface by means of a LevelSet Function and models the obstacles as part of the resolved scales.

5   The IBM is implemented with a three-steps procedure: first, an explicit-in-time forcing is developed based on a novel Ghost-Cell Technique that uses several image points instead of the classical one mirror point. The second step consists in an implicit step projection where the right-hand side of the Poisson equation is modified by means of a Cut-Cell Technique to satisfy the incompressibility constraint. The condition of non-permeability is achieved at the embedded fluid-solid interface by an iterative procedure applied on such modified Poisson equation. In the final step, the turbulent fluxes and the wall model used for Large-Eddy-Simulations (LES) are corrected and a wall model is proposed to ensure consistency of the subgrid scales with the IBM treatment.

   In the second of part of the paper, the IBM is verified and validated for several analytical and benchmark test cases of flows around single bluff bodies with increasing level of complexity. The analysis showed that MNH-IBM reproduces the expected physical features of the flow, which are also found in the atmosphere at much larger scales. Finally the IBM is validated in the

15   LES mode against the Mock Urban Setting (MUST) field experiment, which is characterized by strong roughness caused by the presence of a set of obstacles placed in the atmospheric boundary layer in nearly-neutral stability conditions. The Meso-NH IBM-LES reproduces with reasonable accuracy both the mean flow and turbulent fluctuations observed in this idealized urban environment.

# 1   Introduction

20   Urbanization impacts the physical and dynamical structure of the atmospheric boundary layer, influencing both the local weather and the concentration and residence time of pollutants in the atmosphere, which in turn impact air quality. While the physical mechanisms driving these interactions and their connections to climate change are well understood (the Urban Heat





Island effect, anthropical effects ), their precise quantification remains a major modelling challenge. Accurate predictions of these interactions require modelling and simulating the underlying fluid mechanics processes to resolve the complex terrain featured in large urban areas, including buildings of different sizes, street canyons, parks, etc. For example, it is well known that pollution originates from traffic and industry in and around cities, but the actual dispersion mechanisms are driven by the

local weather. Furthermore, fine-scale flow fluctuations can possibly trigger important nonlinear physicochemical processes and should then be captured by the simulations. The present study addresses these issues focusing on the numerical aspects of the problem.

With the progress in metrology, it is now possible to obtain reliable measurements of the atmospheric conditions over a city. For example, during the Joint Urban experiment (JU2003), scalar dispersion was measured experimentally over the streets

of Oklahoma City (Clawson et al., 2005; Liu et al., 2006). Similarly, the CAPITOUL experiment (2004-2005) conducted in Toulouse, analyzed the turbulent boundary layer developed over the urban topography and evaluated the energy exchanges between surface and atmosphere (Masson et al., 2008; Hidalgo et al., 2008). More recently, the multiscale field study by Allwine et al. (2012) provided meteorological observations and tracer concentrations data in Salt Lake City. Other studies analyzed reduced-scale and/or idealized models to understand urban climate features as in the COSMO (Comprehensive Outdoor Scale

Model Experiment for Urban Climate) project (Moriwaki and Kanda, 2004). For example, Kanda et al. (2007) and Wang et al. (2015) used, respectively, an array of cubic bodies and stones fields as small-scale models.

In order to use in the future these experimental data for model validation, the numerical models need first to be verified for academic test cases and simplified scenarios representative of atmospheric turbulent boundary layer flows. In particular, the flow interaction with buildings or any generic obstacles plays a crucial role in urban flow modelling. The range of scales of

objects acting as obstacles is virtually infinite in urban setting, encompassing large buildings and small vegetation scales and so is the range of the corresponding flow-obstacles interactions. Covering all possible cases is obviously impossible but from a fluid mechanics standpoint one can invoke the principle of similarity which permits, for example, to observe von Kármán streets in the wake of a centimeter-scale cylinder as well as in the cloud layout behind an island. Following this principle, a wide selection of benchmark flows can be analyzed to verify and validate the numerical treatment of fluid-obstacle interaction

with a view to atmospheric applications.

Even if the physical application in our mind is the atmospheric mesoscale reaction to perturbations induced by urban cities, the more the obstacles are considered as a part of the scales numerically resolved the more the results accuracy is. This study presents the development, implementation, verification and validation of an Immersed Boundary Method (IBM) in the Meso-NH model (MNH, Lafore et al., 1998; Lac et al., 2018) for applications to urban flow modelling[1]. This choice was dictated

by the fact that numerical solvers in MNH enforce conservation on structured grids and hence cannot handle body fitted grids with steep topological gradients. The main idea behind IBM is the detection of an interface separating a fluid region (where conservations laws hold) from a solid region (corresponding to the obstacle volume) using different techniques (e.g. markers, LevelSet functions, local volume fraction, etc). As reviewed by Mittal and Iaccarino (2005), two main classes of IBM exist based on the continuous and discrete forcing approaches, respectively. The continuous forcing approach was developed by

---

[1]Meso-NH scientific documentation: http://mesonh.aero.obs-mip.fr/mesonh52/BooksAndGuides



Peskin (1972) for biomechanics applications and consists in the addition of a continuous artificial force (acceleration indeed) in the momentum conservation equation that mimic the effect of the obstacles (heart linings) and drive the flow to relax to no-slip conditions at the wall of the obstacles. This approach and its variant developed by Goldstein et al. (1993) for a rigid interface can suffer from the lack of stiffness (fluid-solid interface is generally spread over few cells) which can be problematic to recover

the boundary layer. Nevertheless, the continuous forcing approaches are very successful in many applications (penalization method as in Angot et al. (1999), fictitious domain method, etc). In the second IBM class, the discrete approach, the boundary conditions are specified at the immersed interface. To simulate flows around non moving and rigid bodies, two sub-classes of discrete approaches can be defined as in Mittal and Iaccarino (2005): direct or indirect approaches, depending on the forcing location (Pierson, 2015). Many types of discrete forcing exist and a non exhaustive list can be: direct forcing in the fluid region

near the interface as in Mohd-Yusof (1997), immersed interface method (Leveque and Li, 1994), Cartesian grid method (Clarke et al., 1986). Depending on how to resolve the partial differential equations, Cartesian grid methods (Ye et al., 1999) are written for finite-volume discretizations (Cut-Cell Technique, CCT) and for finite-difference discretizations (Ghost-Cell Technique, GCT) as in Tseng and Ferziger (2003). Note that the latter technique has been successfully implemented in Weather Research and Forecasting WRF model (Lundquist et al., 2010, 2012).

In this study, a discrete forcing approach is adopted where the fluid-solid interface is modelled by means of a LevelSet function (Sussman et al., 1994). The motivation behind this choice is that we are primality interested in modelling explicitly rigid and non-moving bodies in a turbulent flow, and with sufficiently fine resolution to avoid the large dissipation inherent to the presumed spread interface. Another argument in favor of discrete forcing is that it does not introduce source terms in the conservation equations so that boundary conditions are imposed at the interface and/or in the solid region, the only corrections

to the physical model come from subgrid turbulent parameterizations, and boundary condition is imposed at the interface and/or in the solid region. The idea is that in future mesoscale application, IBM will be used to resolve large obstacles (in the solid region) such as buildings but also mountains, whereas a subgrid drag model will be used to handle unresolved obstacles such as vegetations (Aumond et al., 2013).

The paper is organized as follows. Section 2 briefly describes the general features of MNH. Section 3 details the numerical

implementation of the IBM. Inspired by the works of Bredberg (2000), Piomelli and Balaras (2002), Craft et al. (2002) and to close the turbulence problem, an immersed wall model is proposed in Section 3.3. Sections 4-6 describe the validation of the method for academic flows, respectively potential, inviscid (Lamb, 1932; Batchelor, 2000) and viscous flows (Direct Numerical Simulation). Finally, Section 7 describes IBM-applications to high Reynolds turbulent flows and validation using meteorological data from field experiment. Conclusions are drawn in Section 8.

## 2   The Meso-NH code at a glance

MNH is an atmospheric non hydrostatic research model. Its spatio-temporal resolution is ranging from the large meso-alpha scale (hundred of kilometers and days) down to the micro-scale (meters and seconds). It is massively parallel on nested and structured grids, adapted on most of international hosting computer platforms. Several parametrizations are available: radia-

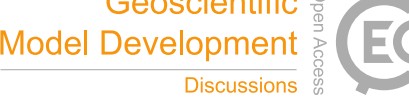

tion, turbulence, microphysics, moist convection with phase change, chemical reactions, electric scheme, externalized surface scheme. In the present study, only two subgrid parametrizations are approached: turbulence and surface schemes.

## 2.1 The conservation laws

The spatial discretization $\mathbf{x}$ is based on the terrain-following coordinates (Gal-Chen and Somerville, 1975). The staggered mesh is regular $\Delta x = \Delta y = \Delta$ in the horizontal directions and a transformation of the vertical one is available in order to fit a non-plane surface. The release of the vertical space step is available where a fine resolution is unnecessary. In the current study, only flat problems are considered with a $\Delta z = \Delta$ restriction for altitudes in presence of immersed obstacles.

The core of the MNH dynamic in its dry version is based on the resolution of the Euler and thermodynamic equations (energy preserving). The anelastic approximation (Lipps and Hemler, 1982; Durran, 1989) is assumed; the reference state is stratified and the density deviation to the hydrostatic case in the buoyancy term is considered. $\rho_r(z)$ and $\theta_r(z)$ are the vertical profiles of the density and potential temperature of the reference state. The system can be simplified into the Boussinesq approximation when considering an uniform reference state. The MNH conservation laws give the tendencies $\frac{\partial}{\partial t}\big|_{law}$. The

prognostic variables are the resolved momentum, the potential temperature and if necessary an arbitrary passive scalar. The prognostic variable is decomposed into a resolved (resp. unresolved) part and an additional prognostic equation on the subgrid turbulent kinetic energy is solved for a Large Eddy Simulation (LES, Sect. 2.3). The potential temperature is defined through the Exner function $\Pi$ and the absolute temperature $T = P/(\rho_r R_d) = \theta(P/P_{r0})^{R_d/C_p} = \theta\Pi$ where $P$ is the local pressure, $P_{r0}$ the reference ground level pressure, $R_d$ the gas constant and $C_p$ the specific heat capacity at pressure constant for dry air. The

thermodynamic equations and an additional passive scalar equation are:

$$\frac{\partial(\rho_r \overline{\theta})}{\partial t}\bigg|_{law} = -\nabla(\rho_r \overline{\theta}\overline{\mathbf{u}}) + \rho_{\mathbf{r}}\overline{\mathbf{f}_\theta} - \nabla.(\rho_{\mathbf{r}}\overline{\theta'\mathbf{u}'}) + \rho_{\mathbf{r}}\overline{\mathbf{F}}_\theta^{\mathbf{\Pi}} \tag{1}$$

$$\frac{\partial(\rho_r \overline{s})}{\partial t}\bigg|_{law} = -\nabla.(\rho_r \overline{s}\overline{\mathbf{u}}) + \rho_{\mathbf{r}}\overline{\mathbf{f}_{\mathbf{r}}} \tag{2}$$

where $r$ corresponds to the subscript of the reference state, $\overline{F}^{\Pi}$ to pressure effects, $\overline{f}$ to other additional terms such as the Coriolis force, molecular diffusion or local source/sink perturbations. The transports of each prognostic scalar in Equations (1), (2) and (6) are made by a Piecewise-Parabolic Method (PPM) with undershoots and overshoots limitation (Colella and Woodward, 1984; Lin and Rood, 1996). The temporal algorithm of the advection term in these scalar transports is a forward-in-time scheme (noted FT). The momentum equations are:

$$\frac{\partial(\rho_r \overline{\mathbf{u}})}{\partial t}\bigg|_{law} = -\nabla.(\rho_r \overline{\mathbf{u}}\otimes\overline{\mathbf{u}}) + \rho_r \overline{\mathbf{f}}_u - \nabla.(\rho_r \overline{\mathbf{u}'\otimes\mathbf{u}'}) + \rho_r \overline{\mathbf{F}}_u^{\Pi} + \rho_r \mathbf{g}\frac{\overline{\theta} - \theta_r}{\theta_r} \tag{3}$$

where $\overline{\mathbf{u}}$ is the resolved wind, $\mathbf{g}$ the acceleration due to the gravity appearing in the buoyancy term, $\nabla.(\rho_r \overline{\mathbf{u}'\otimes\mathbf{u}'})$ the Reynolds stresses. The spatial discretization of $\nabla.(\rho_r \overline{\mathbf{u}}\otimes\overline{\mathbf{u}})$ in Equation (3) can be done by either second- or fourth-order



centered schemes, by either third- or fifth-order Weighted-Essentially-Non-Oscillatory schemes (Jiang and Shu, 1996). The temporal evolution of the resolved wind is achieved by a fourth-order ERK Explicit Runge-Kutta algorithm (Shu and Osher, 1989; Lunet et al., 2017). In the present study $\Delta t$ is fixed to respect the Courant number $\frac{|\overline{\mathbf{u}}^n|\Delta t}{\Delta} < 1$ and no additional time

splitting is implied.

The bottom, lateral wall and top surfaces take a free-slip, impermeable and adiabatic behaviours without the call of an externalized surface scheme. The open boundary condition is a Sommerfeld equation defined as a wave-radiation (Carpenter, 1982) to enforce the large scales and allow the reflection wave damping.

## 2.2   The incompressibility condition

The wind of the resolved scales has to satisfy the continuity equation $\nabla.(\rho_r\overline{\mathbf{u}}^{n+1}) = 0$. The method consists in the projection of the predicted velocity field $\overline{\mathbf{u}}^*$ (solution of Eq. (3) without the pressure term) into the null-divergence subspace. This projection estimates the irrotational correction to apply to $\overline{\mathbf{u}}^*$ through a potential scalar $\Psi^*$:

$$\overline{\mathbf{u}}^{n+1} = \overline{\mathbf{u}}^* - \frac{\Delta t}{\rho_r}\boldsymbol{\nabla}\Psi^* \tag{4}$$

$\Psi^*$ is obtained with the resolution of the pseudo-Poisson equation written as:

$$\nabla.(\rho_r^{-1}\boldsymbol{\nabla}\Psi^*) = \Delta t^{-1}\nabla.\overline{\mathbf{u}}^* \tag{5}$$

The horizontal part of the operator to inverse in the elliptic problem is treated in the Fourier space (Schumann and Sweet, 1988) and its vertical part brings to the classical tridiagonal matrix. The mathematical operator to inverse $\nabla.(\boldsymbol{\nabla})$ is exact

for flat problems (Bernadet, 1995). When the mesh is built with a terrain-following coordinates over a flat surface, the solution of the pressure problem becomes inaccurate. In this orography-presence case, an iterative procedure is employed such as a Richardson, a Conjugate-Gradient (Young and Jea, 1980) or a Residual Conjugate-Gradient (Skamarock et al., 1997) algorithms.

## 2.3   The turbulent subgrid scales

To execute LES, the Reynolds stresses $\nabla.(\rho_r\overline{\mathbf{u}'\otimes\mathbf{u}'})$ appearing in Equation (3) are estimated. The LES closure is done by an eddy-diffusivity approach called 1.5TKE scheme (Cuxart et al., 2000). The isotropic part of the subgrid turbulence is given by the prognosis of the subgrid turbulent kinetic energy $e = \frac{1}{2}(\overline{u'^2} + \overline{v'^2} + \overline{w'^2})$:

$$\left.\frac{\partial(\rho_r e)}{\partial t}\right|_{law} = -\nabla(\rho_r e\overline{\mathbf{u}}) - \rho_r\mathbf{g}\frac{\overline{\theta'\mathbf{u}'}}{\overline{\theta}} - \rho_r\overline{\mathbf{u}'\otimes\mathbf{u}'}.\nabla\overline{\mathbf{u}} + \nabla.(\rho_r K_e l_m\sqrt{e}\nabla e) - \rho_r K_\epsilon e^{3/2}/l_m \tag{6}$$





where $K_e$ and $K_\epsilon$ are constants prescribed in the turbulence scheme, $l_m$ the length scale defining the turbulent viscosity. The dissipation term is directly estimated from $e$ and $l_m$ (the left-hand term in Equation 6). The anisotropic part of the subgrid turbulence is diagnosed from the $\overline{\psi}$ gradient and $e$.

The ground condition can be modelled by the externalized surface scheme SURFEX (Masson et al., 2013) which prescribes the turbulent friction depending on the ground properties as its roughness length for a near-neutral case (cf. Sect. 7.2).

## 3   The IBM forcing in the Meso-NH code

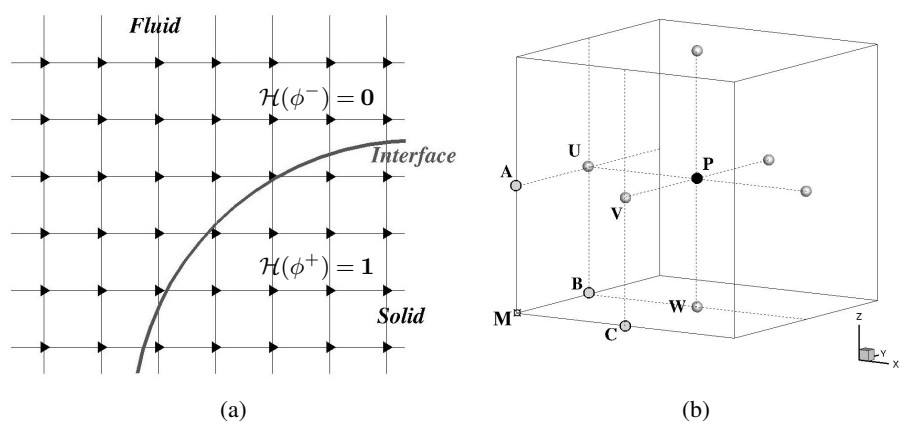

**Figure 1.** *(a) Illustration of an interface separating fluid and solid regions; (b) Definition of the points type per cell: M the geometric/mesh point, P the mass point, U/V/W the velocity nodes and A/B/C the vorticity nodes.*

The numerical domain is divided in two regions: a fluid region where the continuum mechanic equations are acting; a solid region of volume similar to this of the embedded obstacles where these rules become meaningless (Fig. 1-a). After an intensive

comparison with the interface modelling by a local volume fraction function (not shown here), the interface between the fluid and solid regions is modelled by a LSF LevelSet Function (Sussman et al., 1994): $|\phi|$ informs about the minimal distance to the fluid-solid interface and the $\phi$-sign about the region nature: $sgn(\phi) > 0$ for the solid one; otherwise $sgn(\phi) < 0$.

The LSF is estimated at the seven available points type per cell to limit the discretization errors (Fig. 1-b): at the mass point $P$ where prognostic scalar variables are localized, at the three velocity nodes $U/V/W$ where are characterized each projection

$\overline{\mathbf{u}}$, at the A/B/C vorticity nodes employed by turbulent variables. The points of the solid region ($\phi > 0$) acts as external points of the computational grid (as acts external points in a boundary-fitted method at the grid limit). An intensive study had been done (not shown here) to verify the ability of the LSF spatial derivatives to recover the vector normal to the interface and the local





curvature. The forcing based on a GCT Ghost-Cell Technique (resp. CCT Cut-Cell Technique) is applied to the explicit-in-time schemes (resp. the pressure solver) and detailed in Sect. 3.1 (resp. Sect. 3.2).

## 3.1 Ghost-Cell Technique and explicit-in-time schemes

The prognostic variable $\psi$ is decomposed into a resolved (resp. unresolved) part $\overline{\psi}$ (resp. $\psi'$). $\psi' = 0$ in a Direct Numerical Simulation (DNS). $\psi^n$ is the value at the time $n\Delta t$ ($\Delta t$, the time step). The tendencies of the prognostic variables $\overline{\psi} =$

$[\overline{\mathbf{u}}, \overline{\theta}, \overline{s}, (e)]$ can not be deduced from the conservation laws in the solid region. Defining a Heaviside function $\mathcal{H}(\phi)$ and expecting a correction due to IBM where $\phi \geq 0$, a general formulation of the tendencies is written as:

$$\frac{\partial}{\partial t} = \left.\frac{\partial}{\partial t}\right|_{law} + \mathcal{H}(\phi)\left.\frac{\partial}{\partial t}\right|_{ibm} \tag{7}$$

The RHS first term of Equation (7) is given by the conservation laws (Sect. 2.1). $\left.\frac{\partial}{\partial t}\right|_{ibm}$ is the correction of the tendencies

due to the GCT in the solid region and near the immersed interface satisfying the $\overline{\psi}$ desired boundary conditions at $\phi = 0$:

$$\left.\frac{\partial \overline{\psi}}{\partial t}\right|_{ibm} = -\left.\frac{\partial \overline{\psi}}{\partial t}\right|_{law} + \frac{\overline{\psi}^{n+1} - \overline{\psi}^n}{\Delta t} \tag{8}$$

Note that $\left.\frac{\partial \overline{\mathbf{u}}}{\partial t}\right|_{ibm}$ is taken into account in the Explicit-Runge-Kutta (ERK) temporal algorithm. The freeze of the immersed wind conditions in the ERK algorithm had also been tested; it had shown a more unstable behaviour for large Courant number

(not detailed here).

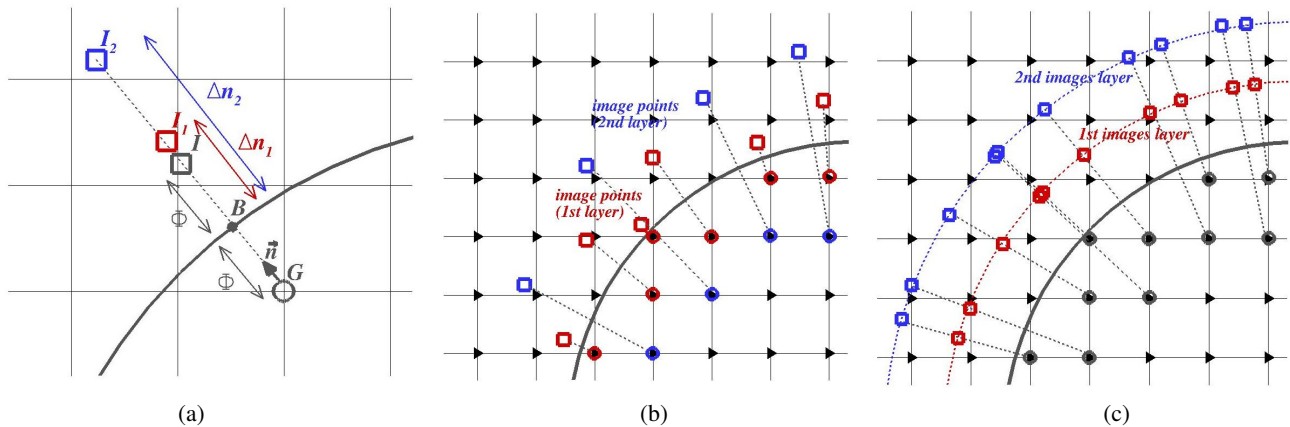

(a)  (b)  (c)

**Figure 2.** *(a) Node definitions acting in the Ghost-Cell Technique : the $G$ ghost , $B$ interface, $\mathbf{n}$ normal vector, $I$ mirror and $(I_1, I_2)$ images. (b resp. c) Illustration of the classical (resp. original) GCT using the mirror (resp. images).*

The forced points are called ghost points and merely renamed ghosts. To estimate the variable $\overline{\psi}$ and for each ghost, the physical information is extracted near the interface and from the fluid region. In the classical GCT (Tseng and Ferziger, 2003)



the fluid information is obtained at a mirror point (noted $I$, merely renamed mirror) found in the normal direction to the interface in such a way that the interface node $B$ is equidistant to $I$ and $G$. Figure 2-a shows the characterization of one ghost $G$, its associated mirror $I$ and the interface node $B$ ($\mathbf{GI} = 2\phi_G\mathbf{n}$,$\mathbf{n}$ the vector normal to the interface). Figure 2-b illustrates several ghosts and mirrors in a two dimensional case. A problematic case regularly met in the mirror interpolation is the vicinity of ghosts with the interface ($\phi_G = -\phi_I << \Delta$, $\Delta$ the space step). The physical information at $I$ is directly related to

the interface's one which can itself be dependent on the fluid information (as it's done in the wall models used in LES). To remove this problem, another way to recover the fluid information is to define image points (noted $I_1$ and $I_2$ in Fig. 2-a, merely renamed images) having a distance to the interface $\Delta$-dependent and not $\phi_G$-dependent: $\mathbf{GI_l} = l\Delta + \phi_G\mathbf{n}$ with $l = (1;2)$. This images characterization is a newly proposed way to recover the fluid information in a GCT.

Figure 2-c illustrates the characterization of this original GCT in a 2D case. The extension to the three dimensional cases

is direct. The triangle symbol corresponds to one of the seven nodes classes (Fig. 1-b) defined per cell in the MNH staggered grid. An intensive comparison of the classical and original GCTs had been done during the studies of the inviscid and viscous flows. The original GCT had given the best results (not detailed here). The original GCT is employed in the rest of this study.

The definition of several images per ghost permits the access to a building of the normal profile of the $\overline{\psi}$ fluid information by an 1D quadratic interpolation (Appendix C). In practice three $I_l$ images are defined for which the location are $\phi_{I_l} = -l\Delta$

with $l = [1/2; 1; 2]$. The forcing thickness depends on the space-order of the numerical schemes. Figure 2-b shows a two-cells thickness forcing (or a two ghost layers). The colour code of the circle symbols illustrates the different ghost layers: red designates the first layer and blue the second one. The characterization of the layer is done by a conditional loop applied direction by direction on the LSF: $\phi(i,j).\phi(i,[j-k_l : j+k_l]) > 0$ and $\phi(i,j).\phi([i-k_l : i+k_l],j) > 0$ ($k_l$ an integer, 2D case). $k_l = 1$ (resp. $k_l = 2$) allows to determine cells truncated by the interface and to define the first (resp. second) layer. Note that the larger

the ghost layer number is the larger the distance between images and ghosts. Such distance implies a computational time cost during parallel exchanges. To limit these triggers, low-order version of some explicit-in-time scheme (such the second-order for the fourth-order centered advection scheme) is computed and employed when $\phi > -\Delta$. The CPU cost of the 'hybrid' advection scheme is largely compensated by the limitation of the ghosts number and parallel exchanges.

The GCT is divided in three main steps: the fluid information recovery, the interface condition and the ghost value.

**The fluid information recovery**. $\overline{\psi}_{I_l}$ for the images contained in a pure fluid cell (all corner nodes are in the fluid region) is recovered by a trilinear interpolation based on Lagrange polynomials (Appendix B). For truncated cells (at least one corner node is in the solid region), $\overline{\psi}_{I_l}$ is recovered using a distance weighting interpolation (Franke, 1982). Several types of such

interpolations are implemented, detailed in Appendix B and tested in Sect. 5. As the boundary condition is expressed in the interface frame and the grid is staggered, the non-collocation of the $\overline{\mathbf{u}}$ components implies firstly to interpolate three different classes of cells (with $U/V/W$ corners, Fig. 1-b) for each $U/V/W$ ghosts, secondly to build the change of frame matrix for which the original GCT presents an interest during the characterization of the direction tangent to the interface (Appendix A).





**The interface condition**. Let $\overline{\psi}_b$ and $\Delta \frac{\partial \overline{\psi}}{\partial n}\big|_b$ the Dirichlet and Neumann conditions on $\overline{\psi}$. The general formulation of the boundary condition $\overline{\psi}(\phi = 0)$ is written as a Robin condition: $\overline{\psi}(\phi = 0) = k_r \overline{\psi}_b + (1 - k_r).(\overline{\psi}(\phi = -l\Delta) - l\Delta \frac{\partial \overline{\psi}}{\partial n}\big|_b)$. Note the $\frac{l\phi}{2}$-approximation on the location of the derivative term. The switch between the Dirichlet condition and the Neumann condition is done through the coefficient $k_r \in [0 : 1]$. To give some examples of Dirichlet condition, $(k_r; \overline{\psi}_b) = (1; 0)$ is imposed: on the $\overline{\mathbf{u}}.\mathbf{n}$ velocity component normal to the interface arising from the impermeability hypothesis, on the $\overline{\mathbf{u}}.\mathbf{t}$ component tangent to the interface for a no-slip hypothesis. To give some examples of Neumann condition imposed by $(k_r; \frac{\partial \overline{\psi}}{\partial n}\big|_b) = (0; 0)$: no flux condition on the potential temperature (as well on a passive scalar, subgrid kinetic energy), free-slip case applied to $\overline{\mathbf{u}}.\mathbf{t}$. Note that the Neumann condition is depending on the chosen image and in practice the selected image $I_l$ is the closest one to the interface ($l = 1/2$).

An interface condition depending on the characteristics of the surrounding fluid such as $\overline{\psi}(\phi = 0) = F(\overline{\psi}_{I_l}; \frac{\partial \overline{\psi}}{\partial n}\big|_{I_l})$ is a wall model. Using two (resp. three) images, simple wall models such as the constant (resp. linear) extrapolation of the $\overline{\psi}$ gradient is reached by the $\frac{\partial^2 \overline{\psi}}{\partial n^2}\big|_{I_l} = 0$ (resp. $\frac{\partial^3 \overline{\psi}}{\partial n^3}\big|_{I_l} = 0$) computation. The consistency between the tangent component to the interface of the resolved wind and the subgrid turbulence is the subject of Section 2.3.

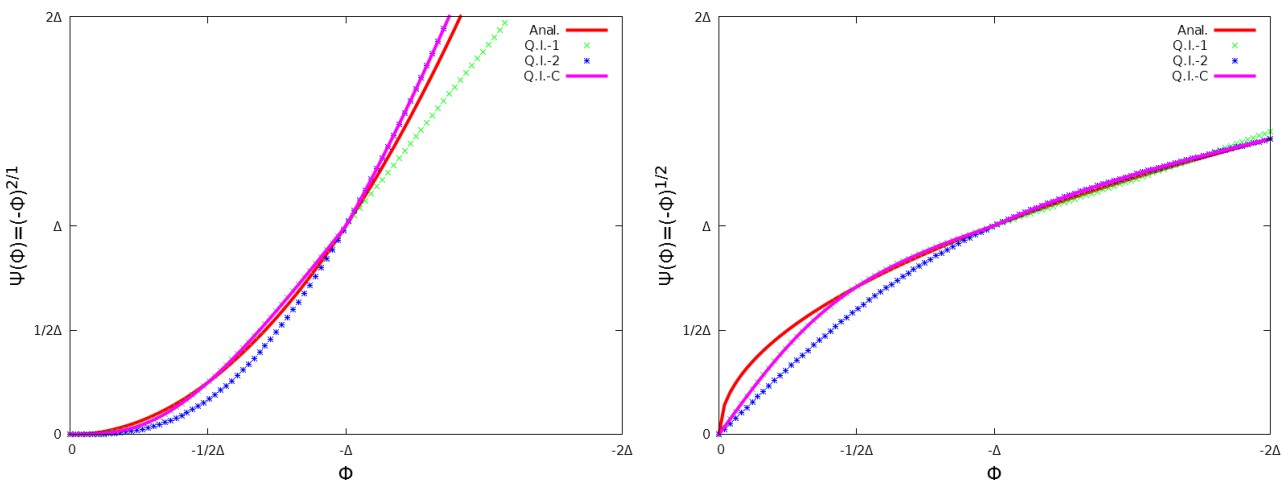

**Figure 3.** *Profile normal to the interface of two fluid informations $\overline{\psi}$: analytical solution (red lines), quadratic interpolation $QI_1$ using $\overline{\psi}_{-\phi=[1/2;1]\Delta}$ (green symbols), $QI_2$ using $\overline{\psi}_{-\phi=[1;2]\Delta}$ (blue symbols), $QI_C$ as a combination of $QI_1$ and $QI_2$ (purple lines).*

**The ghost value**. Knowing $\overline{\psi}(\phi = 0)$ and $\overline{\psi}_{I_l} = \overline{\psi}(\phi = -l\Delta)$, the $\overline{\psi}(\phi^-)$ profile in the fluid region is built by a 1D quadratic interpolation (Appendix C) in the direction normal to the interface. The choice of the images distance to the interface affects the results. To approach at best the expected solution, two quadratic interpolations depending on the used images and one combination of this quadratic interpolations are tested. Figure 3-a and Figure 3-b illustrate these interpolations by considering two analytical profiles (red lines): the quadratic interpolation $QI_1$ (resp. $QI_2$) is based on the images values located at $\phi =$





$1/2\Delta$ and $\phi = \Delta$ and plotted in green symbols (resp. at $\phi = \Delta$ and $\phi = 2\Delta$ plotted in blue symbols). Depending on the analytical profile, Figure 3 shows the influence of the images location choice. As expected, $QI_1$ (resp. $QI_2$) appears to be less accurate than $QI_2$ (resp. $QI_1$) for $\overline{\psi}(\phi \in [-2\Delta : -\Delta])$ (resp. for $\overline{\psi}(\phi \in [-\Delta : 0])$). $QI_C$ is the combination of $QI_1$ and $QI_2$

5 (purple line). $QI_C$ preserves the advantage of each quadratic interpolation and when $\phi_G < \Delta$ (resp. $\phi_G > \Delta$), $QI_1$ (resp. $QI_2$) is used in the rest of the study.

Knowing $\overline{\psi}^{n+1}(\phi^-)$ at the end of the MNH temporal loop with $QI_C$, the $\overline{\psi}^{n+1}(\phi^+)$ profile is extrapolated from the fluid region to the solid region by applying an anti-symmetry $\overline{\psi}^{n+1}(\phi^+) = 2\overline{\psi}^{n+1}(0) - \overline{\psi}^{n+1}(\phi^-)$. The ghost value is estimated and the $\overline{\psi}$-gradient at the interface is also recovered.

10 ## 3.2 Cut-Cell Technique and pressure solver

First looking at the RHS of (5), the $\frac{\partial(\rho_r \overline{\mathbf{u}}^*)}{\partial t}\big|_{law}$ coming from the resolution of the explicit-in-time schemes near the interface and in the solid regions badly affects the $\nabla.\overline{\mathbf{u}}^*$ computation (note that the GCT operates after the step projection). Therefore the fictive wind of the solid region can spread errors in the fluid region during the pressure resolution. To avoid it, a correction of the pressure solver is proposed.

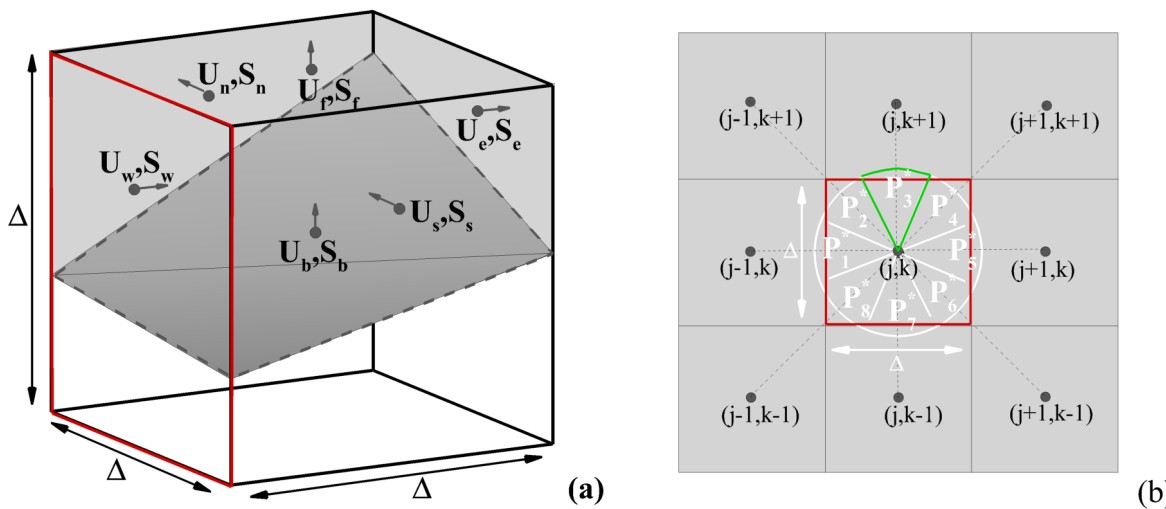

**Figure 4.** *(a) Momentum fluxes balance for an arbitrary truncated cell of volume $\mathcal{V}$ where the $\overline{u_i}^*$ velocities ($U_i$ in the figure, $i = [e, w, n, s, b, f]$) are supported by the $\mathcal{S}_i$ surfaces in grey colour; the transparent volume is a part of the solid body. (b) Segmentation of the $\mathcal{S}_w$ arbitrary surface (red border) in eigth $P_p^*$ pieces of cake (the border of $P_3^*$ is indicated in green).*

The elliptic problem (5) is re-written as a resolution of the linear system $\mathcal{P}.\Psi^* = \mathcal{Q}$. In the standard MNH version, $\nabla.\overline{\mathbf{u}}^* = \mathcal{Q}$ is estimated using a finite difference approach. To uncouple the solid region from the fluid region our revisited version enforces





a null-divergence for pure solid cells and estimates the balance of momentum fluxes by a finite volume approach for truncated cells (noted $\mathcal{Q}_{cct}$):

$$
5 \quad \mathcal{V}\nabla.\overline{\mathbf{u}}^* = \int_{\mathcal{V}_f} \nabla.\overline{\mathbf{u}}^* d\mathcal{V} + \int_{\mathcal{V}_s} \nabla\overline{\mathbf{u}}^* d\mathcal{V} = \sum \pm \overline{u_i}^*.\mathcal{S}_i = \sum \pm \widetilde{\Delta^2\overline{u_i}}^* \quad (9)
$$

where $\mathcal{V} = \Delta^3$ is the cell volume, $\mathcal{V}_f$ (resp. $\mathcal{V}_s$) the fluid (resp. solid) part of $\mathcal{V}$, $\mathcal{S}_i$ the cell surfaces where $i$ is the index of each surface orientation [e,w,n,s,b,f] as it illustrates in Figure 4-a.

According to the Green-Ostrogradski theorem, the $\overline{u_i}^*\mathcal{S}_i$ calculation is the classical way of a CCT Cut-Cell Technique (Yang
10  et al., 1997) to estimate the velocity divergence. A similar approach is here performed re-building the flux $\widetilde{\Delta^2\overline{u}}^*$ for truncated and solid cells.

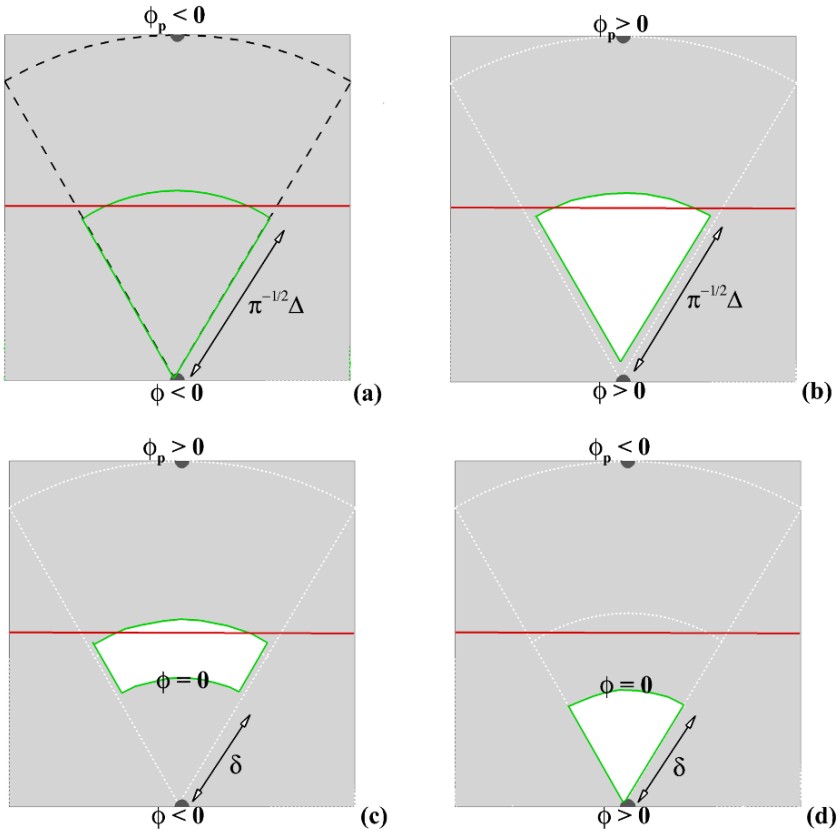

**Figure 5.** *(a,b,c,d) $\pm\widetilde{\Delta^2\overline{u_i}}^*$ calculations depending on the signs of $\phi_i = (\phi)$ and $\phi_p$ on an arbitrary piece of cake. The white (resp. grey) region corresponds to the solid (resp. fluid) one of $P_p^*$ (same colour code as in Fig. 4).*




The $\pm\widetilde{\Delta^2 \overline{u_i}^*}$ calculation consists in a weighting of the out-fluxes and in-fluxes function of the fluid and cell surfaces ratio (Fig. 4-a). Figure 4-b gives an example of the west surface ($i = w$, red border) where $\widetilde{\Delta^2 \overline{u_w}^*}(j,k)$ is calculated using the LSF value $\phi = \phi_w$ and the ones of the eight adjacent nodes $\phi_p(j \pm 1, k \pm 1)$. A disk of radius $\sqrt[-1]{\pi}\Delta$ is split in eight 'piece of cake' segments $P_p^*$ ($p = [1:8]$). A LSF linear interpolation detects or not the interface location. In presence of an interface, its distance from the studied node is $0 < \delta_p < \sqrt[-1]{\pi}\Delta$. Knowing $\delta_p$, the momentum fluxes balance is formulated for a non-moving body as ($p$ is the index of the 'piece of cake' and i the index of the cell surface):

$$\widetilde{\Delta^2 \overline{u_i}^*} = \frac{\Delta^2}{8}[\sum_{p=1}^{8}\mathcal{H}(-\phi_p)\mathcal{H}(-\phi_i)\overline{\mathbf{u_i}}^* + \sum_{p=1}^{8}\mathcal{H}(-\phi_p\phi_i).|\mathcal{H}(-\phi_p) - \pi(\frac{\delta_p}{\Delta})^2|.(\mathcal{H}(-\phi_p)\overline{\mathbf{u}}_p^* + \mathcal{H}(-\phi_i)\overline{\mathbf{u_i}}^*)] \tag{10}$$

The four encountered cases correspond to a pure fluid cell $\widetilde{\Delta^2 \overline{u_i}^*} = \frac{\Delta^2}{8}\sum_{p=1}^{8}\overline{\mathbf{u_i}}^*$ when $\phi_p < 0$ and $\phi_i < 0$ (Fig. 5-a); a pure solid cell $\widetilde{\Delta^2 \overline{u_i}^*} = 0$ when $\phi_p > 0$ and $\phi_i > 0$ (Fig. 5-b); two types of truncated cells depending on the fluid/solid nature of the main node for which $\phi_p.\phi_i < 0$ (Fig. 5-c/d). Using Eq. (10), Equations (9) are solved and lead to the RHS computation of (5).

Knowing $\mathcal{Q}_{cct}$, the reflection concerns now the $\mathcal{P}$ matrix to inverse. The classical interface condition on the potential $\Psi^*$ is a homogeneous Neumann condition $\frac{\partial\Psi^*}{\partial\phi} = 0$. Using a Boundary Fitted Method (BFM), the interface condition of the moving or non-moving body (Auguste, 2010) appears only on the border of a numerical domain. Using an IBM and without any impact of this interface condition on the $\mathcal{P}$-coefficients, the impermeability character of solid obstacles is not achieved. Due to the inversion of the horizontal part of $\mathcal{P}$ by a Fast Fourier Transform (Schumann and Sweet, 1988), the solution of calculating $\mathcal{P}_{cct}$ appears to be problematic. The adopted solution consists in an iterative procedure as used in MNH for non-flat problems. The non-respect of the $\Psi^*$-condition in $\mathcal{P}$ leads to an unwell posed system and the iterative procedure goes to spread to the entire fluid domain the enforcement of the null-divergence imposed on solid cells. The resolution of the pseudo-Poisson equation (5) brings to $\Psi^* \rightarrow \Psi^{*M} = \sum_{m=1}^{M}\mathcal{P}^{-1}.\mathcal{Q}_{cct}^m$ where $M$ is the number of iterations. This number is limited by a convergence criterion (compromise between incompressibility satisfaction and CPU cost). Many iterative procedures are available in MNH and originally developed for non Cartesian grids. A Richardson and a Preconditioned Conjugate-Residual algorithms had been here adapted to the obstacles immersion. The newly modified pressure solver is tested and validated in Sect. 4.

## 3.3 Consistency with the turbulence scheme

The turbulent characteristics are highly affected by a surface interaction. As a consequence and for LES, the subgrid turbulence scheme (Sect. 2.3) is modified in presence of immersed obstacles on the subgrid turbulent kinetic energy equation, mixing length computation and Reynolds stresses diagnosis.

**The Subgrid Turbulent Kinetic Energy condition**. The explicit-in-time resolution of Eq.(6) claims a GCT forcing and an interface condition on the STKE $e$. Commonly, the $e-$profile is considered parabolically in the viscous sublayer (Craft et al., 2002; Bredberg, 2000) and constant in the inertial and wake/outer layers (Kalitzin et al., 2005; Capizzano, 2011). Due to the high turbulent Reynolds number $Re_t \approx \mathcal{O}(10^4 - 10^5)$ encountered, a homogeneous Neumann condition is applied at the





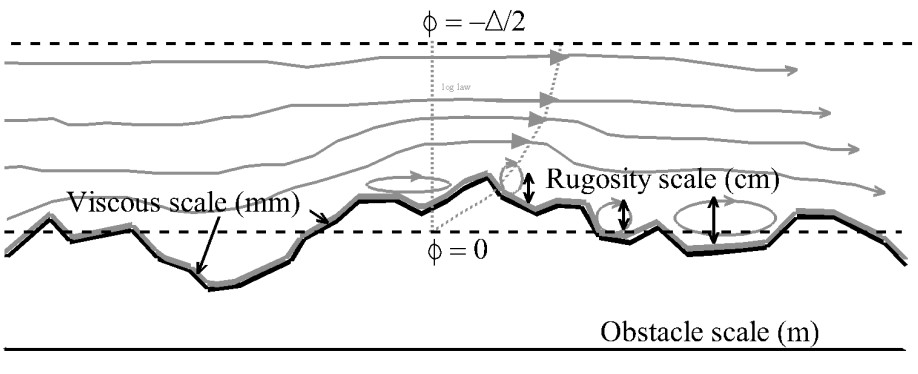

**Figure 6.** *Illustration of the unresolved physical processes near a non-idealized solid wall (black line) in an atmospheric context: the length scale based on the viscous effects (grey line) is drastically smaller than the roughness length. The roughness length approaches the scale of smallest eddies and governs the log-law profile.*

immersed interface. The equilibrium between production and dissipation of STKE could be discussed and controverted; this choice acts as a first stage in the IBM development.

**The near-wall correction of the mixing length**. The von Kármán limitation due to immersed walls acts through the LSF and the upper limit on the mixing length $l_m$ near interface becomes $min(kz, -\phi, \Delta)$ with a banning of negative values in the solid region. Whatever the production of subgrid turbulent kinetic energy and the turbulent shear, the lower limit $l_m(-\phi \leq 0)$ induces a null value of the diagnosed surface fluxes. In addition, a singularity appears in the dissipative term $\rho_r K_\epsilon e^{3/2}/l_m$ of Equation (6). By a pragmatic reasoning, the singularity due to $l_m^{-1}(\phi \to 0^-) \to \infty$ amounts to say that modelled length scales are smaller than the Kolmogorov scale $(\nu^3 \epsilon^{-1})^{\frac{1}{4}}$. In other words and by considering the unrealistic hypothesis that the Kolmogorov scale is achieved, the turbulence should vanish. In order to overcome this unwell posed-problem, a $l_m$ lower limit has to be specified. In the study of atmospheric flows around buildings, a characteristic thickness of the viscous layer $H/\sqrt{Re}$ can be defined around a $H$ bluff body for a Reynolds number based on the obstacle scale ($H \approx \mathcal{O}(10m); Re \approx \mathcal{O}(10^7)$. This thickness estimate is also proportional to $E\nu/u^*$ ($E \approx 9.8$ is commonly employed) where the friction velocity $u^*$ is about the centimeter per second. Following these estimates, the length scale due to the viscous effects $z_\nu^{ib}$ belongs to the millimetres domain in the expected atmospheric cases. Looking after a building surface and its large heterogeneity (door, windows, surface characteristics), its roughness length $z_0^{ib}$ is at least in the decimeter domain and $z_0^{ib} > z_\nu^{ib}$ (Illustration in Fig. 6). For 'fluid mechanics' application (weaker Re) and smooth surfaces, $z_\nu^{ib} > z_0^{ib}$ could be encountered.

Finally, we assume that $z^{ib} = max(z_0^{ib}, z_\nu^{ib})$ and that $z^{ib}$ is related to the size of smallest unresolved eddies near walls (dissipative scale). The mixing length near wall is $z^{ib} < l_m < min(kz, -\phi, \Delta)$.



**The turbulent fluxes correction**. The $\overline{\psi}$-gradient and the turbulent diffusion $\mathcal{O}(z^{ib}\sqrt{e})$ prescribe the turbulent fluxes at the immersed interfaces. As a first step in the MNH-IBM implementation, no-flux condition on the mean potential temperature is imposed bringing to a zero-value of the sensible heat flux. Writing the normal and tangent parts of the mean velocity field near an interface as $\overline{\mathbf{u}} = \overline{u_t}\mathbf{t} + \overline{u_n}\mathbf{n}$ (Appendix A), $\widetilde{u_t}$ is needed as interface condition of $\overline{u_t}$ to recover a gradient consistent with the turbulent shear.

Considering the space resolution sufficiently far from the dissipation scale $\Delta > z^{ib}$, the Prandtl (1925) or Kármán (1930) logarithmic profile is assumed in the vicinity of the wall according to $\widetilde{u_t} = \frac{u^*}{k}ln(1 + \frac{\Delta}{z^{ib}})$.

Considering $\Delta$ as the limit of the resolved scales, most of the TKE is contained in the subgrid TKE (STKE) when $-\phi < \Delta$ and $K_{tke}\sqrt{e} \sim \sqrt{TKE}$ with a constant $K_{tke} \gtrsim 1$ (the classic $k$ notation is kept for the von Kármán constant). This assumption is reinforced by the homogeneous Neumann condition applied on $e$. This approach derives from the RANS (Reynolds-Averaged Navier Stokes) approaches and the velocity friction is formulated as $u^* = K_{tke}\sqrt[4]{C_\mu}\sqrt{e}$ where $C_\mu$ is a constant evolving between 0.03 (atmospheric applications) and 0.09 (fluid mechanics applications). Adding a damping function for the viscous cases (low turbulent Reynolds number, $Re_t < 20$), the tangent wall velocity is written as:

$$\widetilde{u_t} = \frac{K_{tke}\sqrt[4]{C_\mu}\sqrt{e(\phi = -\Delta/2)}}{k}ln(1 + \frac{\Delta}{z^{ib}}(1 - exp(-\frac{20\Delta}{z_\nu^{ib}}))) \tag{11}$$

Finally the pragmatic limitation $\widetilde{u_t} \leq \overline{u_t}(\phi = -\Delta/2)$ operates if the subgrid TKE value is too high. The proposed dynamic wall-model evolves between the no-slip and free-slip conditions. If the subgrid turbulence is weak or if the physical problem is fully resolved, the viscous layer is well-modelled and $\overline{u_t}(\phi = 0) \rightarrow 0$. Otherwise for an intense subgrid turbulence or a fully unresolved problem, the shear due to the wall presence is not perceived and $\frac{d\overline{u_t}}{d\phi}\big|_{\phi=0} \rightarrow 0$. In the numerical practice, the wall-model establishes an equilibrium between the production of subgrid TKE and the mean parietal friction. This immersed wall-model is a first step. It raises issues as the validity of the log-law near a singularity (sharp edges or corners). Nevertheless, Sections 7.1 and 7.2 show LES results employing this proposition. After numerical investigations (not shown here), $K_{tke}\sqrt[4]{C_\mu} \approx 1$ appears as a suitable choice.

## 4  Potential flows

Isolated from the rest of the code, the resolution of the pseudo-Poisson equation (5) leads to potential solutions (Sect. 2.2). Theoretical ones are available for flow developed around a non-deformable obstacle such as an infinite cylinder or a sphere (Milne-Thomson, 1968; Batchelor, 2000). The two bodies are here investigated. The flow around the infinite cylinder is predominantly presented.

Figure 7 illustrates the cylinder case. The fluid density is considered as constant in time and in space. The flow is initially imposed as spatially homogeneous with a constant module of velocity $U_\infty$ and parallel streamlines (Fig. 7-a). This initialization does not respect the conservation of the momentum flux and the irrotationnal correction of the projection method goes to





recover this conservation. In the same time, the impermeability of the cylinder of diameter $D_{cyl} = 2R_{cyl}$ is achieved. Figure 7-b shows the streamlines obtained with the MNH pressure solver modified to take into account the presence of immersed obstacles (Sect. 3.2). Defining $\mathbf{x}$ as the direction parallel to the initial streamlines and $\mathbf{y}$ as the perpendicular one, the expected solution is $\mathbf{u}.U_\infty^{-1} = cos\alpha(1 - \frac{R_{cyl}^2}{r^2})\mathbf{x} - sin\alpha(1 + \frac{R_{cyl}^2}{r^2})\mathbf{y}$ (single and non-confined body, $(\alpha; r)$ cylindrical coordinates). The numerical confinement is comment hereafter, characterized by $L = L_{cyl}/R_{cyl}$ where $L_{cyl}$ is the distance separating the lateral domain surfaces (Fig. 7-a).

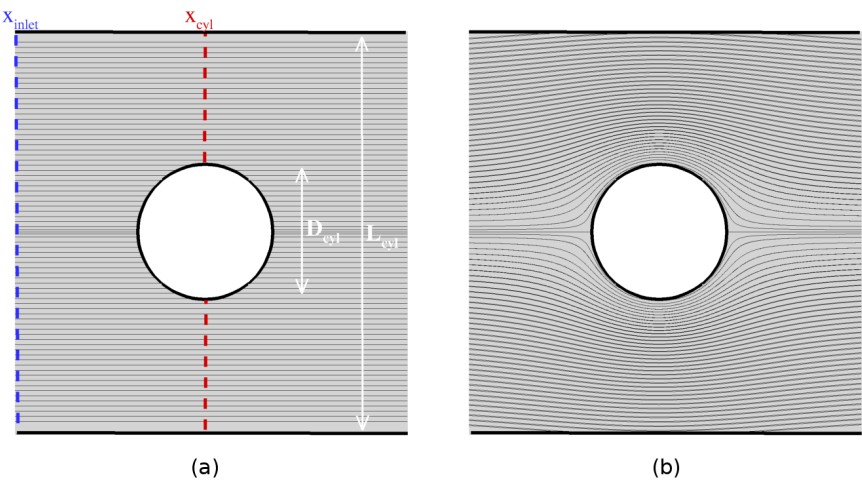

**Figure 7.** *Potential flow around a cylinder: (a) initial state around the body of diameter $D_{cyl} = 2R_{cyl}$; (b) streamlines obtained after the Poisson equation resolution. The confinement is defined as $L = L_{cyl}/R_{cyl}$).*

The RICH Richardson and the RESI Residual Conjugate Gradient iterative methods are tested (Sect. 3.2). Figure 8-a plots the evolution of the dimensionless residue $R(k)$ (based on a characteristic divergence $U_\infty/\Delta$) with the iterations number $k$ and obtained with the confinement $L = 16$. The two algorithms converge with a weak dependence to the spatial discretizations ($N = [4(red); 8(green); 16(blue); 32(purple)]$ nodes per $R_{cyl}$). $\frac{dR(k)}{dk}(RESI) \lesssim 3\frac{dR(k)}{dk}(RICH)$, so RESI demonstrates its highest velocity convergence. Even if RICH is about $20\%$ faster per iteration than RESI, the global CPU cost of the last one is lowest for a same solver residue $R(k)$. For this reason and due to an a priori higher radius convergence, RESI is adopted. Note that the momentum flux computed after the solver convergence at the $x_{cyl}$ location (Fig. 7-a) shows a good mass preserving with a relative error of $[0.48\%(N = 4); 0.20\%(N = 8); 0.18\%(N = 16); 0.14\%(N = 32)]$ in regard of the incoming flux localized by its $x_{inlet}$ longitudinal coordinate. Similar results had been obtained with a spherical body (not shown here).

With a change of Galilean reference frame, this study corresponds to an uniform body acceleration $\mathbf{a_b}$ in a fluid initially at rest. However a possible viscous term, the hydrodynamic force exerted on the body, is reduced to the added mass effect $\mathcal{A}m_f\mathbf{a_b}$ for $\Delta t \to 0$. $\mathcal{A}$ is the dimensionless coefficient and $m_f$ the displaced fluid mass. $\mathcal{A}_{cyl}$ theoretically equals 1 in the





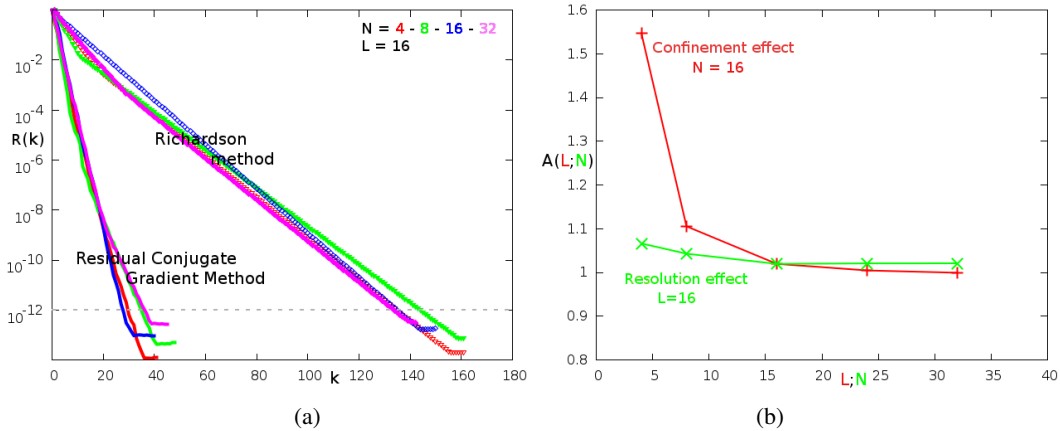

(a)  (b)

**Figure 8.** *Potential flow around a cylinder: (a) Velocity convergence of two iterative methods (Residual Conjugate Gradient, Richardson) for different spatial resolutions $N = [4:32](L = 16)$; (b) Evolution of the added mass coefficient $A(N;L)$ with the confinement $L = L_{cyl}/R_{cyl}$ ($N = 16$) and with the nodes number per radius cylinder $N$ ($L = 16$). The confinement is defined in Fig. 8-a.*

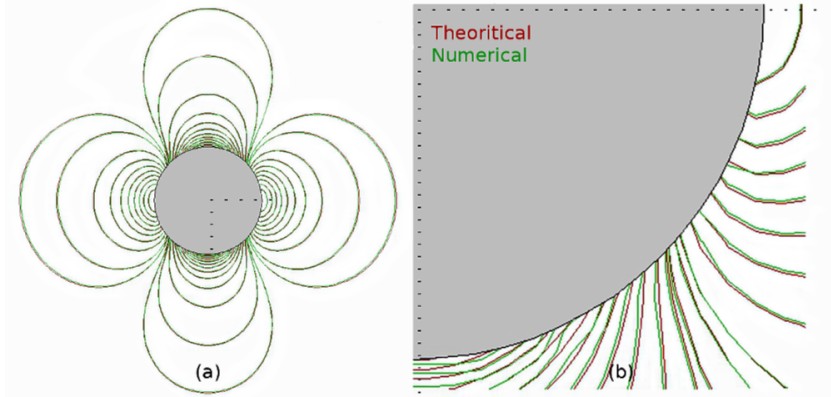

**Figure 9.** *Potential solution around a sphere: (a) kinetic energy in an arbitrary symmetry plane; (b) zoom.*

non-confined cylinder case (Lamb, 1932). The red curve of Figure 8-b illustrates the effect of the confinement $L$ on $\mathcal{A}_{cyl}$ for a $N = 16$ resolution. Unsurprisingly $\mathcal{A}_{cyl}$ increases with the confinement. The weak dependence of $\mathcal{A}_{cyl}$ with $L > 16$ allows to consider the body as isolated for $L \sim 16$. The green curve of Figure 8-b shows the impact of the space resolution for $L = 16$. The numerical added mass coefficient is in good agreement with the theoretical one presenting a relative error of about 2% for

5  $N > 16$. It induces a well-respect of the impermeability hypothesis at the immersed interface.

A similar study for a spherical body gives $\mathcal{A}_{sph} = \frac{1}{2} + 0.4\%$. Figure 9 illustrates the contours of the kinetic energy around the sphere in an arbitrary symmetry plane. The green contours (numerical solutions) fit well with the red contours (theoretical solutions).





The Taylor vortices are investigated (Fig. 10-a) imposing in the RHS of Equation (5) the divergence $\nabla.(\overline{\mathbf{u}}^*) = -\pi(l^2 + m^2)cos(\pi lx)sin(\pi my)$ where $l = m = cste$. The error norms ($L_p = \sqrt[p]{\sum |P_n - P_t|^p}$ where $P_n$ is the numerical pressure and $P_t$ the theoretical one) are estimated in presence or not of an immersed cylindrical body (Fig. 10). The space second-order of the pressure solver is recovered without IBM. The order decreases with IBM and stays consistent regarding the $L_{p=(\infty;1;2)}$

slopes. Note that an immersed square or sphere give similar results (not shown here).

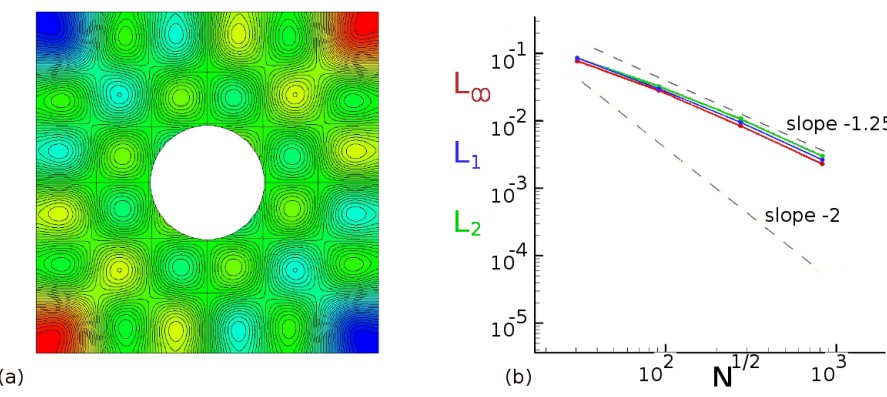

**Figure 10.** *Taylor vortices around a cylinder: (a) illustration; (b) $L_\infty$, $L_1$ and $L_2$ norms function of the space resolution.*

Finally, the irrotationnal solution around two 2D Agnesi hills (or 'bells') is investigated with IBM and the Boundary-Fitted Method (BFM, terrain-following coordinates). The topography is characterized by a height $h_a$ and a shape $h(x) = \frac{h_a}{1+(\frac{k_a \cdot x}{h_a})^2}$ ($k_a = 4$, bell 1; $k_a = 8$, bell 2). The bells slope is here arbitrarily and respectively described as gentle or steep. Figure 11 shows the pressure contours obtained with IBM (left) and BFM (right) for a gentle (top) and a steep (bottom) shape. The minimal

pressure value is localized at the top of each bell and goes to zero far from this location. The reference BFM and IBM simulations with the fine resolution ($N = 160$ nodes per $h_a$, red colour) show a good agreement for each hill. The blue (resp. green) colour corresponds to a coarser mesh employing $N/3$ (resp. $N/9$) nodes per $h_a$. Weak differences appear between the $N$ and $N/3$ meshes for both IBM and BFM revealing a good space convergence (Fig. 11-a/b). Numerical errors are visible with IBM near the interface but the Venturi effect is well-modelled. Differences become more significant with the $N/9$ mesh especially

with the BFM-BELL2 presenting the highest curvature value (Fig. 11-d).

To conclude, this section validates the modification to the pressure solver. IBM appears less accurate than BFM when the ground presents low curvature in regard of the space resolution. It seems more pertinent than BFM to model high interfaces such as sharp edges or corners.





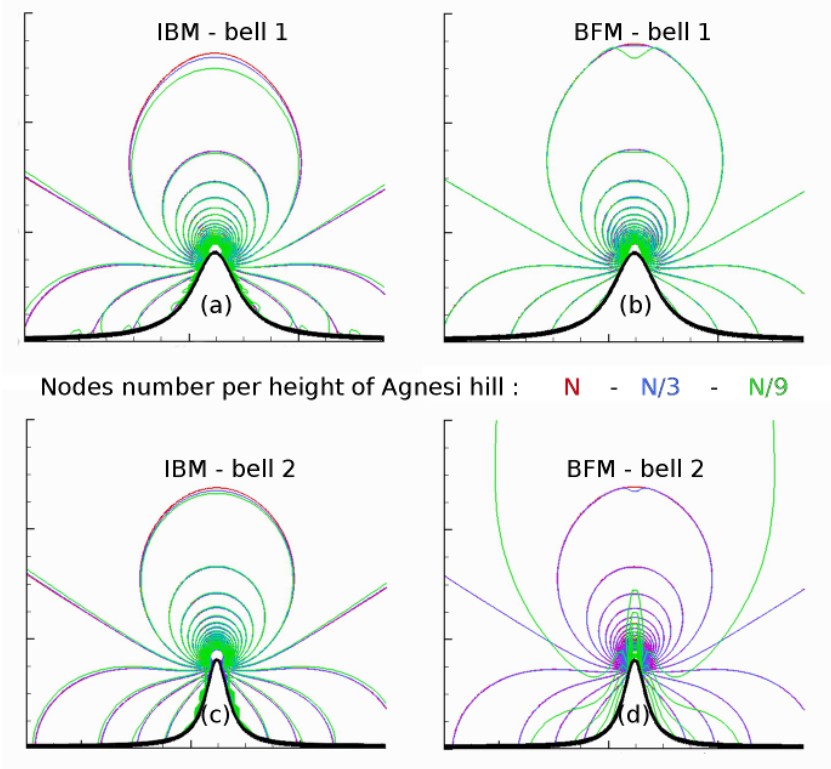

**Figure 11.** *Potential flow function of the space resolution (colour code) plotting the pressure contours around two bells (top: BELL1, gentle slope; bottom: BELL2, steep slope) and obtained with IBM (left) and the boundary-fitted method (right).*

## 5 Inviscid flows

For most atmospheric applications, the region size where the fluid molecular viscosity $\nu_f$ influences the dynamic is sufficiently small to be considered as negligible (Sect. 2.3). Solving the Euler equations, the impact of the numerical diffusion could be significant especially near the fluid-solid interface. The adopted strategy with IBM is to model the advection term with a low-order scheme near the interface (Sect. 3). The WEN3 third-order Weight-Essential-Non-Oscillatory and CEN2 second-order centered schemes are available in MNH. Far from the interface a CEN4 fourth-order centered scheme is employed.

The vorticity equation for a 2D inviscid flow reveals no production in time. Solving the Euler equations, the numerical vorticity production at the immersed surface of a cylindrical body is here studied initializing the simulation with the potential solution. To fit as well the potential solution, a non-trivial condition is employed on the tangent velocity $\frac{\partial^3 \overline{u}_t}{\partial n^3} = 0$. Expecting a numerical vorticity sufficiently controlled to avoid the flow separation, the effect of the artificial diffusion $\nu_{art}\Delta\overline{\mathbf{u}}$ injected with CEN2 is compared to the WEN3 intrinsically diffusive behaviour. Furthermore this study estimates the 3D interpolations impact (Appendix B).





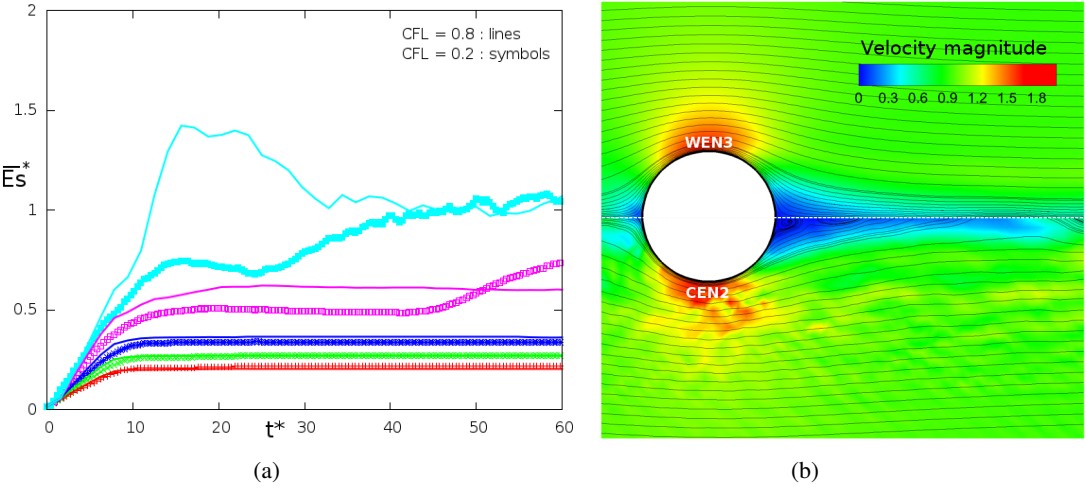

**Figure 12.** *Solving the Euler equations: (a) Vorticity production $\overline{E}_s^*(t^*)$ influenced by the Courant number (CFL=0.8, line; CFL=0.2, symbol) and by the $\nu_{art}$ artificial viscosity ([red; green; blue; purple; cyan] respectively $\nu_{art} = \nu_{art}^{ref}[1; 2; 4; 16; 256]^{-1}$) using an advection CEN2 second-order centered scheme near the interface; (b) Velocity magnitude field obtained with a WEN3 third-order Weight-Essential-Non-Oscillatory scheme (top) and CEN2+$\nu_{art}^{ref}256^{-1}$ (bottom). CEN4 is the advection scheme used far from the interface. The mesh is the coarser one (MESH1).*

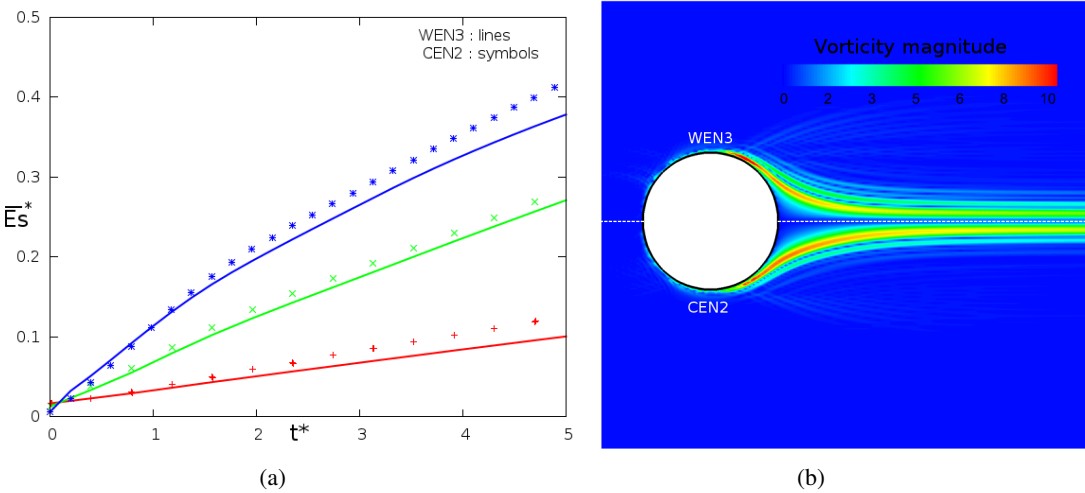

**Figure 13.** *Solving the Euler equations: (a) Influence of the space resolution (red: MESH1; green: MESH2; blue: MESH3) on the vorticity production $\overline{E}_s^*(t^*)$ when the near-interface advection is modelled by WEN3 (line) and CEN2+$\nu_{art}^{ref}$ (symbol); (b) Vorticity magnitude field obtained by WEN3 (top) and CEN2+$\nu_{art}^{ref}$ (bottom).*





Figure 12-a plots the evolution in time of the enstrophy $\overline{E}_s^*(t^*) = \frac{D_{cyl}}{U_\infty \mathcal{V}_f} \int_{\mathcal{V}_f} |\nabla \times \overline{\mathbf{u}}| d\mathcal{V}$ depending on the $\Delta t$ time step and $\nu_{art}$ using CEN2 near the interface ($U_\infty$ the velocity of the incoming flow, $\mathcal{V}_f$ the integration volume in the fluid region). The enstrophy increases in time and reaches a mean value when the produced vorticity near the interface is evacuated from the numerical domain and in the body wake. Except the simulations with low artificial diffusion (symbols/curves in cyan/purple colours), the vorticity production is weakly dependent of the physical time and CFL Courant number (symbols/curves in red/green/blue colours). It induces $\nu_{art}$ proportional to $U_\infty \Delta x \approx \mathcal{O}(\frac{\Delta x^2 CFL}{\Delta t})$. A reference value of the artificial viscosity is also defined as $\nu_{art}^{ref} = \frac{\Delta x^2}{\Delta t} CFL$.

Figure 12-b illustrates the vorticity field in the vicinity of the interface between the intrinsically diffusive WEN3 and CEN2+$\nu_{art}$ with $\nu_{art} = \nu_{art}^{ref} 256^{-1}$. The streamlines are maintained without detachment near the interface with WEN3. Otherwise the CEN2 solution with low artificial diffusion presents numerical instabilities and vortex shedding.

Figure 13-a plots the enstrophy evolution for three meshes (colour code) for WEN3 (lines) and CEN2+$\nu_{art}$ with $\nu_{art} = \nu_{art}^{ref}$ (symbols). MESH1 (10 nodes per $D_{cyl}$), MESH2 (20 nodes per $D_{cyl}$) and MESH3 (40 nodes per $D_{cyl}$) are respectively the coarse, intermediate and fine mesh. The border of the numerical domain is always distant from the cylinder of more than $10R_{cyl}$. The CEN2+$\nu_{art}^{ref}$ vorticity production appears fairly close to the WEN3's one for the three space resolutions. Figure 13-b corroborates the last comment presenting the vorticity contours dimensionless by $\frac{D}{U_\infty}$. A suitable $\nu_{art}$ combined with CEN2 choice is also in the range of the too diffusive WEN3 results and the growth of numerical instabilities. CEN2+$\nu_{art}^{ref} 4^{-1}$ is retained as the advection scheme of the mean wind near an immersed interface.

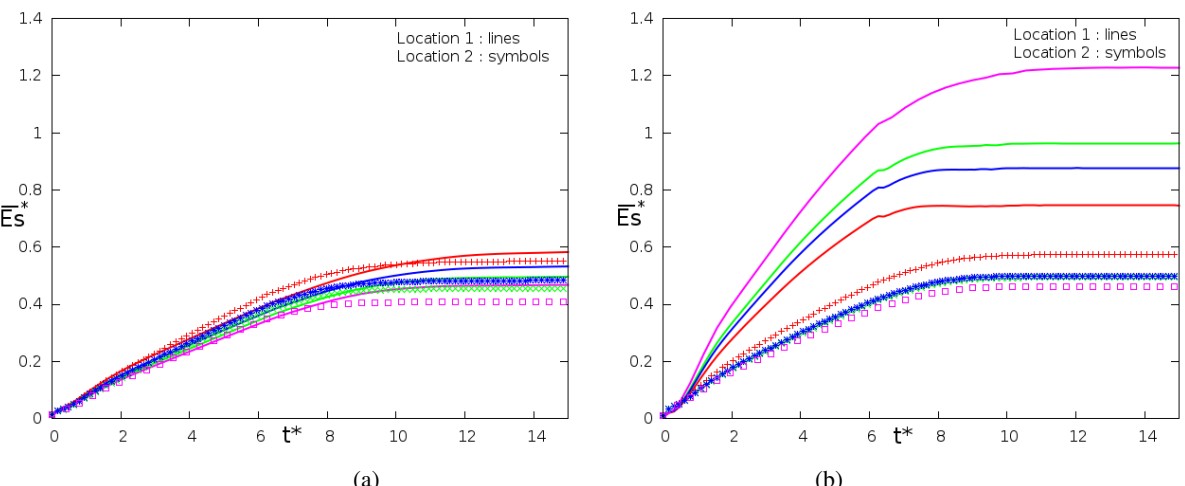

(a)  (b)

**Figure 14.** *Influence of the inverse distance weighting interpolations employed near the interface (red: IDW1; green: IDW2; blue: MDW1; purple: MDW2) and depending on the body center location (line: geometric node; symbol: mass node): (a resp. b) Time evolution of the enstrophy $\overline{E}_s^*(t^*)$ obtained with the intermediate MESH2 (resp. fine MESH3).*





The vorticity production is expected independent of the discretization. Up to now the geometric center of the cylinder was placed on a mass node (Fig. 1-b, P point here referenced as location 1). The previous results are completed with those obtained when the body center is placed on a mesh node (Fig. 1-b, M point here referenced as location 2). The additional simulations are executed on MESH2 and MESH3 for two 3D interpolations types near the interface (Appendix B): the Inverse Distance Weighting (IDW$\alpha$), the Modified Distance Weighting (MDW$\alpha$). These interpolations are compared varying the $\alpha$-exponent in the formulations (B5). In any case the Lagrange interpolation is used far from the interface. Figure 14-a (resp. b) shows the $\overline{E}_s^*(t^*)$ time evolution for MESH2 (resp. MESH3). The colour code corresponds to the 3D-interpolations type. IDW1 gives the results the more independent of the body location and space resolution. The formulation (B5) with $\alpha = 1$ is retained as the 3D interpolation in the vicinity of the interface.

## 6  Resolution of the Navier-Stokes equations: DNS

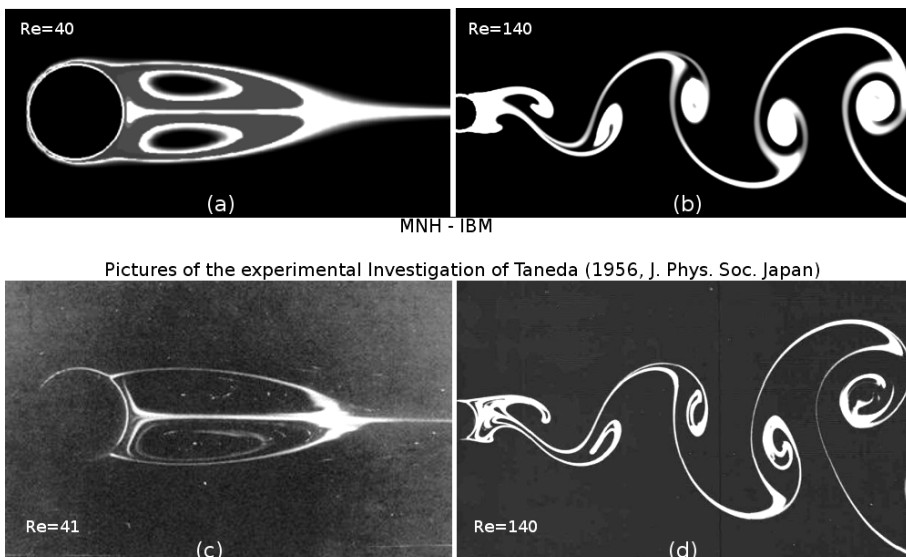

**Figure 15.** *Eddies structure in a viscous fluid: Steady (left, $Re \approx 40$) and unsteady (right, $Re \approx 140$) solutions obtained by the current numerical investigation (top, MNH-IBM and $30 pts/D_{cyl}$) and by the famous experimental Taneda (1956) investigation (bottom).*
*The visualization is due to the presence of passive tracer injected on the body surface and transported by the flow.*

A molecular diffusion ($\nu_f$, the kinematic viscosity) is explicitly added as a source term in Eq. (3) to achieve a converged in time and in space solution of a non-linear problem. The Navier-Stokes equations are then solved by DNS. A pure dynamic and well-documented case which naturally follows the previous ones is studied here. This physical case is the wake past a circular cylinder (non-stratified flow) at two moderate Reynolds numbers $Re = (40; 140)$. One of the forerunners is Taneda (1956) who experimentally studied the nature of the eddies structure.





| Authors | $\theta_d$ (°) | $l_r/D_{cyl}$ | $b/D_{cyl}$ | $a/D_{cyl}$ |
|---|---|---|---|---|
| Coutanceau and Bouard (1977) | 53.8 | 2.13 | 0.76 | 0.59 |
| Linnick and Fasel (2005) | 53.6 | 2.28 | 0.72 | 0.60 |
| Taira and Colonius (2007) | 53.7 | 2.30 | 0.73 | 0.60 |
| Bouchon et al. (2012) | 53.4 | 2.26 | 0.71 | 0.60 |
| Gautier et al. (2013) | 53.6 | 2.24 | 0.71 | 0.59 |
| MNH-IBM | $\approx 54$ | $\approx 2.2$ | $\approx 0.7$ | $\approx 0.6$ |

**Table 1.** *Description of the standing eddies in the wake of the solid cylinder ($Re = 40$): comparison of the separation angle $\theta_d$ (°), recirculating length $l_r$ (m) and vortex core location $(a; b)$ between the literature and MNH-IBM.*

15    Taneda (1956) found a regular Hopf bifurcation at a critical Reynolds number $Re_c = \frac{U_\infty D_{cyl}}{\nu_f} \approx 45$. Below $Re_c$ and above $Re > 5$, a boundary layer separation brings to a steady recirculating region in the near-wake (Fig. 15-c). Above $Re_c \approx 45$, an unsteady mode breaks the planar symmetry and the body wake presents an alternate vortex shedding (Fig. 15-d). The standing eddy (resp. the von Kármán street) obtained by MNH-IBM at $Re = 40$ (resp. 140) is visualized in Figure 15-a (resp. b) by the injection of a passive tracer on the body surface.

The standing eddies at $Re < Re_c$ are commonly described with a $\theta_d$ detachment angle, $l_r$ recirculating length and $(a; b)$
location of the vortex core (Fig. 16). The limit of the numerical domain is $10D_{cyl}$ upstream the obstacle for the inlet condition ($U_\infty$, the uniform incoming velocity) and lateral condition (slip condition), $15D_{cyl}$ for the outlet condition allowing the vorticity evacuation. Three regular Cartesian meshes are built with $10/20/30$ nodes per $D_{cyl}$.

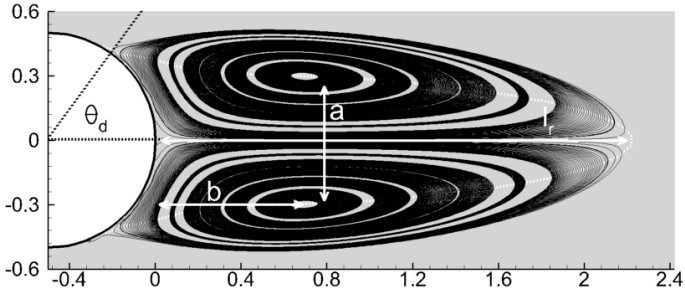

**Figure 16.** *Recirculating region at $Re = 40$ (MNH-IBM, $20pts/D_{cyl}$): definition of the $\theta_d$ (°) separation angle, $l_r$ recirculating length and $(a; b)$ vortex core location. The distance are dimensionless by $D_{cyl}$.*

The $20pts/D_{cyl}$ and $30pts/D_{cyl}$ meshes present a good spatial convergence and weak differences at $Re = 40$. $\theta_d \sim 53°\pm2°$
and the recirculation length $l_r/D_{cyl} \sim 2.2 \pm0.05$. The $10pts/D_{cyl}$ mesh shows more discrepancies which are attributable to the non-ability of the coarsest resolution to capture the viscous boundary layer for which the thickness evolves in $D_{cyl} \sqrt[-1]{Re}$. Table 1 compares the $20pts/D_{cyl}$ results with a part of the results literature collected in Gautier et al. (2013).



The focus is on the unsteady mode at $Re = 140$. The ratio between the characteristic time of inertial effects $D_{cyl}/U_\infty$ and the one related to the vortex shedding $1/f$ defines the Strouhal number $St = \frac{fD}{U_\infty}$. Brazza et al. (1986), Park et al. (1998) and Stalberg et al. (2006) comfort the equation $St(Re) = -3,3265/Re + 0,1816 + 1,6.10^{-4}/Re$ proposed by Williamson (1989). MNH-IBM obtains $St(Re = 140) \in [0.177 : 0.179]$ and an absolute maximum relative error lower than $2\%$ in regard of the Williamson (1989) formulation with the two finer resolutions.

This study validates the MNH-IBM DNS dedicated to the viscous flows past a bluff body at moderate Reynolds number.

## 7 Resolution of the Navier-Stokes equations: LES

This section is devoted to turbulent flows approaching our perspective: the simulation of an atmospheric flow over a city. The turbulent flows around a cubic body vertically confined in a channel and over an urban-like roughness (set of obstacles) are here described. MNH-IBM is explicitly compared to experimental investigations in the two cases. The comparisons to other LES of the literature will be mentioned.

*Common hypothesis and methods.* The fluid is considered as neutrally stratified. The Coriolis term is negligible due to the addressed space and time scales. The turbulent diffusion is modelled by the subgrid TKE 1.5 scheme transported by PPM (Sect. 2.3). All surfaces are considered as non-permeable and the IBM wall-model (Sect. 3.3) is activated. A $(x, y, z)$ reference frame is defined (z, vertical direction) and the velocity vector is written as $\mathbf{u(t)} = u(t)\mathbf{x} + v(t)\mathbf{y} + w(t)\mathbf{z}$. A time simulation is needed after to establish the turbulence state (not shown here). The over-line notation refers to the mean value in time in this section.

### 7.1 Flow over a surface mounted-cube

Using static pressure measurements, laser-sheet and oil-film visualizations, Martinuzzi and Tropea (1993) and Hussein and Martinuzzi (1996) had provided an important data bank contribution describing the dynamic developed around a cubic body placed in a channel (Fig. 17-a). RANS and LES had explored in detail this physical case (Breuer et al., 1996; Shah and Ferziger, 1997; Rodi et al., 1997; Frank, 1999; Krajnovic and Davidson, 2002; Farhadi and Rahnama, 2006).

*Physical details.* A cube ($H$ side) is placed in a channel of $2H$ height. The channel is sufficiently large in the span-wise direction to consider the cube as single in that direction. Turbulent flow is generated in the channel upstream the cube with a mean bulk velocity $U_b$. Defining the dimensionless wall coordinate $z^+ = u^*.z/\nu_f$, the stream-wise upstream velocity corresponds to a log-law for smooth walls $\overline{u}(z^+).u^{*-1} = 5.54 + \frac{1}{\kappa}log(z^+)$ as described in Hussein and Martinuzzi (1996). The Reynolds number defined by the mean bulk velocity, the cube height and the molecular diffusion is $Re \approx 40000$.

The mean flow around the cube presents a set of five recirculating regions (Fig. 17-b). Each cube surface is associated with one of these regions: the $A/B$ vortex separations in front of the cube which spread laterally in a horseshoe $D$, two vortices near side-walls $E$, one $F$ on the roof and main arch vortex $G$ downstream.

*Numerical details.* The top and bottom surfaces, the cubic body are modelled by the IBM. A small value of the roughness length $z_0/H \approx 10^{-6}$ is imposed (low value to model a smooth interface, viscous scale intervention in the $z^+$ calculation). $x$



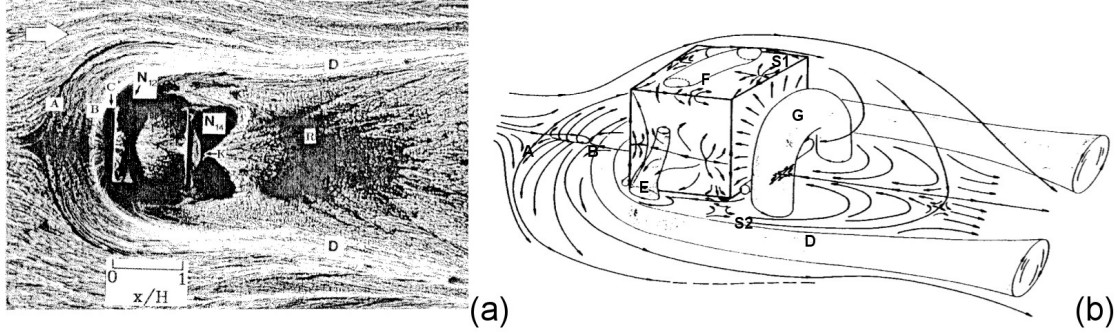

**Figure 17.** *(a) Top visualization of the flow around a cube (Hussein and Martinuzzi, 1996); (b) Schematic representation of the time-average vortex structure around a cube (Martinuzzi and Tropea, 1993) and index of the recirculating regions.*

(resp. $y$) is the stream-wise (resp. span-wise) direction. The size of the grid is set as $(x, y, z) = (-24H : 8H, -4H : 4H, 0H : 2H)$ with a location of the cube center at $(\frac{H}{2}; \frac{H}{2}; \frac{H}{2})$. Three regular Cartesian meshes are employed with a respective space step $H/\Delta = [10; 20; 40]$. $x/H \in [-24 : -4]$ is a region employed to model the fully turbulent character of the incoming flow. The incoming turbulent state is obtained by the IBM and a pseudo-recycling method inspired from the works of Lund (1998), Mayor et al. (2002) and Yang and Meneveau (2016) (not detailed here). The vertical profiles of the stream-wise velocity and turbulent intensity $\sqrt{\overline{u'^2}}/U_b^2 \approx 2.10^{-2}$ (Hussein and Martinuzzi, 1996) are recovered at $x/H \sim -4$. We mention that the turbulence generation should deserve more details but we prefer only to concentrate our comments in the cube wake.

*Results.* Figures 18-(a-b-c) show the time-average streamlines in the vertical symmetry plane of the cube obtained by MNH-IBM for the three space resolutions. The streamlines of the coarse, medium and fine resolution are respectively in red, green and blue colour. The discretization order of the fine resolution is close to that of most literature's LES except Shah and Ferziger (1997) who had used a far more precise grid near wall. The same figure obtained by the experimental investigation is given in Figure 18-d. The size of the front (resp. rear) region is characterized by the recirculating length $x_f/H$ (resp . $x_r/H$). The experiment gives $x_f/H \approx [1.04 : 1.05]$ and $x_r/H \approx [1.64 : 1.67]$. The literature's LES give the ranges $x_f/H \in [0.81 : 1.28]$ and $x_r/H \in [1.38 : 2.25]$. For the two finest resolutions (green and blue colour), the overall prediction of MNH-IBM recovers a consistent mean topologic structure. MNH-IBM obtains for the two finest resolutions $x_f/H \in [0.99 : 1.21]$ and $x_r/H \in [1.48 : 1.55]$. MNH-IBM does not capture as most of LES the two dividing lines $A/B$ commented by Martinuzzi and Tropea (1993) but only captures a flattened vortex. However the bifurcation point near the rear edge and ground is not detected by MNH-IBM while this point was modelled in Rodi et al. (1997). This bifurcation point was also commented in Martinuzzi and Tropea (1993) even if their experimental uncertainty did not allow to visualize it in Fig. 18-d.

Figure 18-e shows an MNH-IBM instantaneous flow field with the Q-criterion (Hunt et al., 1988) and as the literature mentions, it presents a highly intermittent character clearly visible with the quasi-disappearance of the $D$ horseshoe and $G$ arch. A frequency $f$ of vortex shedding dominates the highly-intermittent activity in the body wake bringing to the experimental Strouhal number $St = \frac{f.H}{U_b} \approx 0.145$. A MNH-IBM energy spectrum (Discrete Fourier Transform of $w(t)$ in the body wake)





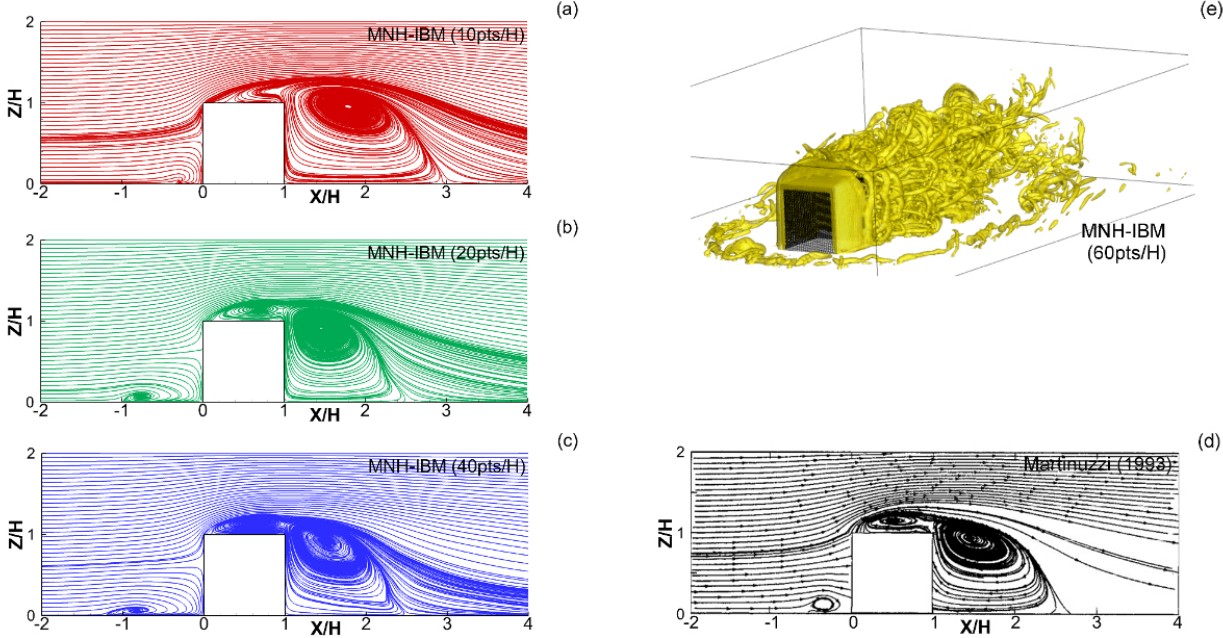

**Figure 18.** *Vertical symmetry plane of the mean flow: (a-b-c) MNH-IBM time-average streamlines; (d) Streamlines observed by (Martinuzzi and Tropea, 1993); (e) MNH-IBM instantaneous visualization of the Q-criterion (Hunt et al., 1988).*

find a peak $St \in [0.10 : 0.12]$ for all the studied resolutions and a $\sim -5/3$ energetic cascade slope for larger wave numbers (not shown). The $St$ values obtained by MNH-IBM are slightly less than the experimental one but stays consistent with the

$St \in [0.10 : 0.15]$ range of other LES.

Still in the vertical symmetry plane of the mean flow, Figure 19 plots at four longitudinal positions $x/H = (1/2; 1; 2; 4)$ the vertical profiles of time-averaged quantities related to the steady ($\overline{u}$, $\overline{w}$) and unsteady (TKE, $\overline{u'w'}$) parts of the solution. The colour code corresponds to the spatial resolution ($10pts/H$ in red; $20pts/H$ in green; $40pts/H$ in blue). Note that the $U_b$ bulk velocity was set to the unity and therefore presented variables in Figure 19 are dimensionless. In most of the sub-figures, the

higher space resolution is the lower the gap with the experiment is. The top of Figure 19 brings to a similar conclusion with that of the literature's LES: the stream-wise velocity is well-recovered. The counterflow at the rear and roof of the cube near $(x/H; z/H) \approx (1; 1)$ informs about the existence of the bifurcation point $S1$ (Fig. 19-b). An underestimation appears on the counterflow at $(x/H; z/H) \approx (2; 0)$ and is frequently observed in the literature (Fig. 19-c). Some discrepancies are found on two $\overline{w}$-profiles (Fig. 19-f/g). Note that Shah and Ferziger (1997) and Krajnovic and Davidson (2002) highlight the difficulty

to recover it. The turbulent kinetic energy and the Reynolds stress are correctly predicted at the cube roof (Fig. 19-i/j). The vertical profiles of TKE and $\overline{u'w'}$ downstream the cube show an overestimate tendency (Fig. 19-k/p/l). No experimental data is available on TKE at $x/H = 4$ (Fig. 19-l) but using the $\overline{u'^2}(x/H = 4)$ experimental value (not shown here) and the $\overline{u'^2}$/TKE



**Figure 19.** *Mean vertical profiles of velocities (top), turbulent kinetic energy and Reynolds stress (bottom). The lines corresponds to the MNH-IBM results. The symbols are the Martinuzzi and Tropea (1993) data except for TKE (Hussein and Martinuzzi, 1996). The profiles are given at four longitudinal locations: (a-e-i-m) $x/H = 1/2$; (b-f-j-n) $x/H = 1$; (c-g-k-o) $x/H = 2$; (d-h-l-p) $x/H = 4$.*





ratio obtained by MNH-IBM, we suspect that TKE($x/H = 4$) is still overestimated. This turbulence diagnosis are relatively similar to those of Farhadi and Rahnama (2006).

*Sensitivity study.* Some tests have shown a sensitivity of the solution with both the incoming turbulence and constants in the immersed wall-model. Indeed the existence of the small vortex near the saddle node $S1$ (Depardon et al., 2006) turns out to be strongly dependent of the inlet turbulence (the less the turbulent intensity is the bigger the size of this vortex is); the reattachment or not of the roof vortex $F$ with the main arch $G$ is also observed. This sensitivity follows the observations of
Castro and Robins (1977) who studied the cube placed in a uniform or turbulent incoming flow. In the same way the existence of the vortex near the saddle node $S2$ (Frank, 1999) is dependent of the ground boundary condition. The $u^* \overset{-1}{\sqrt{e}} = K_{tke} \sqrt[1/4]{C_\mu}$ ratio and the $z_0$ roughness length fixed in the immersed wall-model (Sect. 3.3) impact the nature of the incoming turbulent state for which the surface shear plays an important role. To give a significant example and if a null-value of $K_{tke}$ is applied on the channel surfaces (non-slip condition), $x_f$ highly increases and the vortex at $S2$ appears. Otherwise $K_{tke}$ and $z_0$ do not
crucially affect the nature of the dynamic in the vicinity of the cube surfaces; the pressure gradient between front and back faces governs.

To conclude this section and despite the uncertainties of the inlet condition, MNH-IBM is in a good agreement with the experiments of Martinuzzi and Tropea (1993), Hussein and Martinuzzi (1996) and other LES using a resolution of about forty points per cube length. Even if the coarsest resolution looses a part of the expected physics, it maintains a suitable modelling of
the largest structures of the flow. A parametric study on the inlet turbulence generation has led to fix $K_{tke} \approx 2$ in Equation (11).

## 7.2    The Mock Urban Setting Test experiment (MUST)

The MUST is an experimental campaign organized during the early Autumn 2001 in the Utah's West desert (Biltoft, 2001, 2002). Its objective was to quantify the dispersion of a passive tracer (propylene) in a dry atmospheric context over a topography reproducing a near-urban canopy. The main interest lies in the similitude between this experiment and a pollution episode due
to a toxic gas propagation over a city with high population densities. It provides extensive measurements of meteorological variables and scalar dispersion informations.

*Physical details.* The near-regular array is composed of 120 containers. Figure 20 gives a photograph and a picture of the topography. The containers are equivalent in volume and shape. Their spatial dimensions are $(L_x, L_y, L_z) = \sim (2.4, 12.2, 2.5)m$ and the horizontal distance between containers is $\mathcal{O}(10)m$. Following the table II in Yee and Biltoft (2004), the 2681829 case
(the 25/09/2001 at 18h29) is selected. The Monin-Obukhov length is $L_{mo} \approx 28000m$ and the stability condition is supposed neutral; the buoyancy effects and the sensible heat flux are negligible in regard of the inertia effects and the turbulent shear. The incoming flow shows a mean horizontal angle with the containers layout (green arrow, Fig. 20-b). The MNH-IBM results are compared to the experimental measurements reachable at several altitudes (4m, 8m, 16m) at a S (South) tower and at the main T tower placed in the array centre. The towers location is indicated in Fig. 20-b. A roughness length $z_0 = 0.045m$ is given by
the experience and related to the surrounding desert vegetation.

*Numerical details.* The externalized scheme SURFEX (Masson et al., 2013) models the ground friction. LSF is generated to represent the topography and IBM is used to model the containers. The smallest characteristic container length is discretized





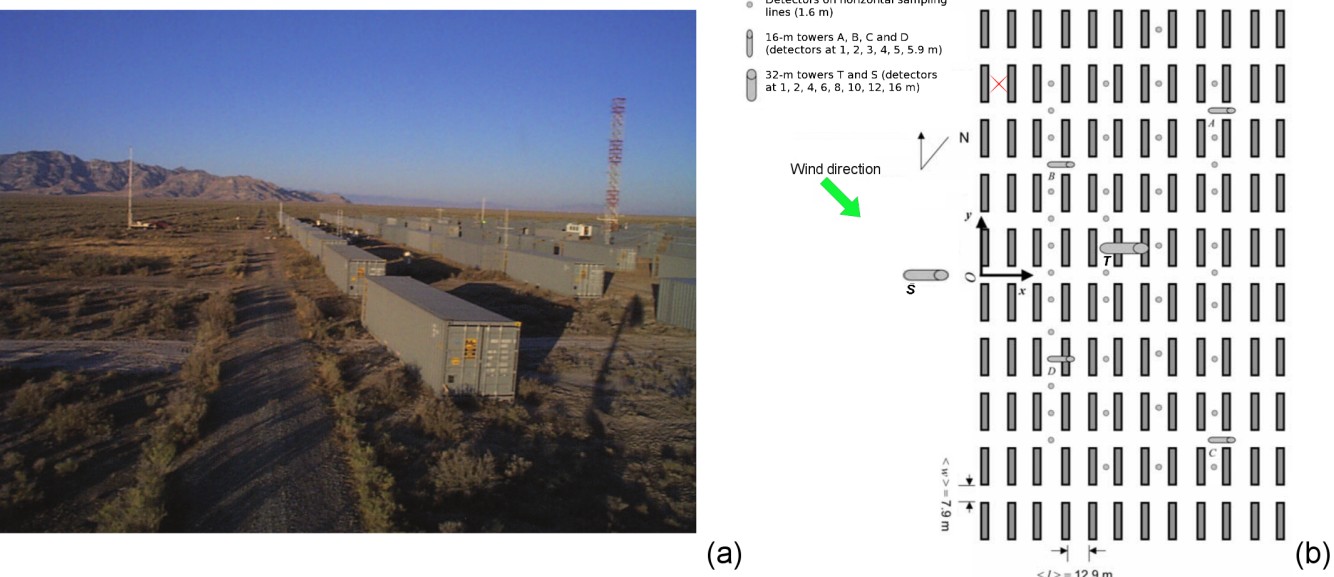

(a)                         (b)

**Figure 20.** *(a) Photograph of the MUST containers in the Utah's West desert; (b) Schematic representation of the containers layout. The location of the concentration (detectors) and wind sensors (S and T towers) are indicated, as well as the position (red cross) of the pollutant release and the direction of the incoming flow (green arrow) of the case 2681829. Source: Biltoft (2001) and Yee and Biltoft (2004).*

by $\mathcal{O}(10)$ cells. The mesh is a Cartesian and regular grid for $z < 10m$ with a space step $\Delta = 0.2m$. Above ten meters, the vertical space step is released in geometric progression of 1.08 ratio. The altitude of the numerical domain top is about 8 times the container height.

The distance between the horizontal limit of the computational domain and the array is about 20 times the container height. The large scale flow is forced by an open boundary condition. A mean horizontal angle of -41° with the $x$ direction is fixed for all altitudes; note that the low angle deviation and the turbulence observed upstream the containers in the experiment are not

numerically considered. Following the experimental data given at the S tower (Fig. 21-a, blue symbols) and assuming a log-law $|\overline{\mathbf{u}}| = \frac{u^*}{\kappa} log(1. + z/z_0)$, a least-square regression estimates a friction velocity $u^* = 0.71 m.s^{-1}$ and allows to build the vertical profile of the mean incoming flow (Fig. 21-a, blue line). This $u^*$ value is comforted by the experimental one $u^*_{exp} = 0.68 m.s^{-1}$ found by a sonic anemometer at the feet of the S tower. The same inlet condition is used in the numerical studies of Hanna et al. (2004), Milliez and Carissimo (2007) and Donnelly et al. (2009). Note that an additional term $\mathcal{O}(L_{mo}^{-1})$ appears in their

formulation but is negligible in the 2681829 case.

*Results.* The black line (MNH-IBM) and symbols (Yee and Biltoft, 2004) of Figure 21-a (resp -b) show the impact of the near-urban canopy on the vertical profile of the $|\overline{\mathbf{u}}|$ (resp. wind angle) in the core of the array (T tower). Not surprisingly the canopy induces a global slowdown of $|\overline{\mathbf{u}}|$ near the ground and up to $z \lesssim 8m$. A decrease of the mean horizontal wind angle is found at the same altitudes. This deviation is related to the containers orientation which tends the flow to be aligned with





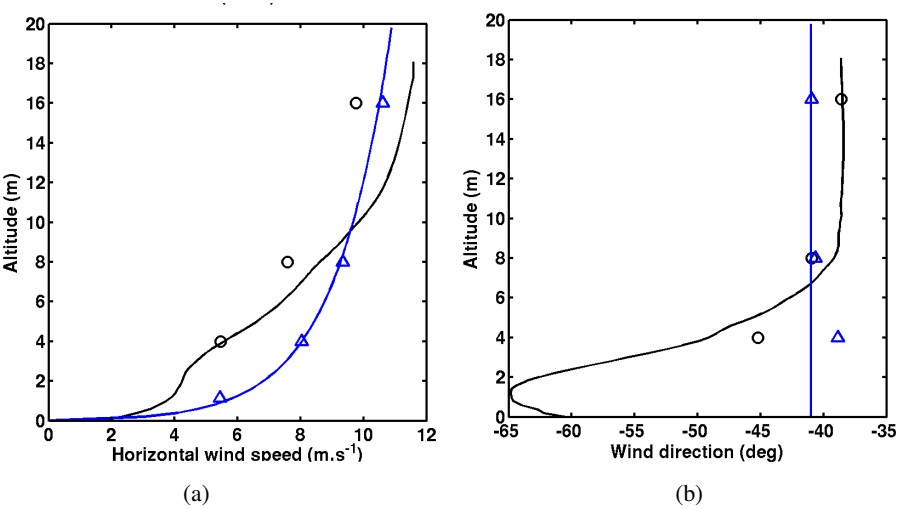

**Figure 21.** *Mean vertical profiles in the MUST experiment: (a) $|\overline{\mathbf{u}}|$ horizontal velocity; (b) $atan(\overline{v}/\overline{u})$ wind direction at the S (blue) and T (black) towers. The symbols (resp. lines) are the Yee and Biltoft (2004) experimental measurements (resp. MNH-IBM results).*

the $y$ direction. The same pattern is discussed in Milliez and Carissimo (2007). A wind acceleration and an increase of the wind angle are observed by MNH-IBM for $z \gtrsim 8m$ as in LES results of Konig (2013) and Dejoan et al. (2010) on similar MUST cases. This few degrees deviation is observed by the experiment but not the acceleration. A part of this acceleration may be explained by a Venturi effect and a too closed top limit of the computational domain (numerical confinement, under investigations).

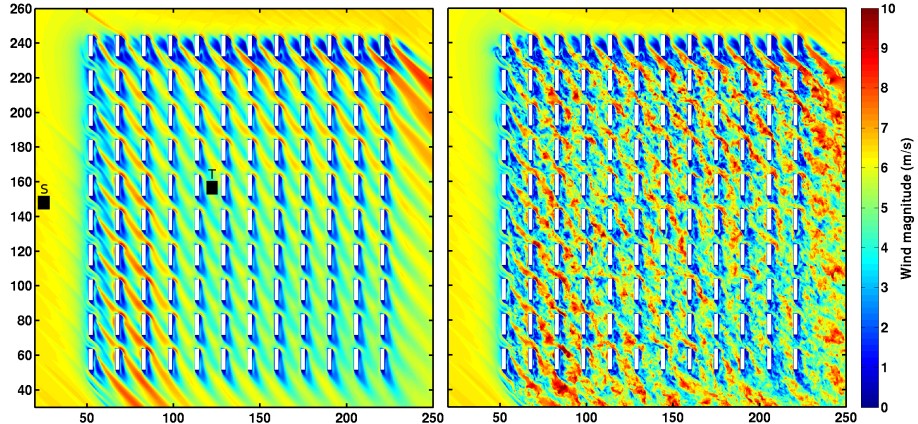

**Figure 22.** *Wind at the horizontal cut at $z = 1.6m$: (a) Time-averaged on 200s; (b) Instantaneous at $t = 200s$. The black squares indicate the location of the T and S towers (MNH-IBM results)*

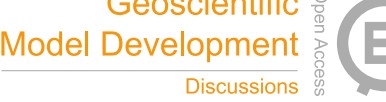

Figure 22-a (resp. Fig. 22-b) illustrates the time-averaged horizontal wind field $|\overline{\mathbf{u}}|$ (resp. $|\mathbf{u}|$ at $t = 200s$) at the 1.6m altitude. These figures highlight the fact that the incoming flow is not turbulent in the simulation. This assumed gap constitutes a perspective (Camelli et al., 2005) not directly linked to IBM. It allows also to introduce here some comments on the turbulence state. The atmospheric turbulence is dependent of the roughness length ($z_0$ considered as constant due to the homogeneous and flat desert over few miles upstream). The first container rows are the scene of the boundary layer transition and act as a region of strong roughness change. The turbulence observed in the urban-like canopy has two origins: the incoming turbulence and that of induced by the containers presence. The contribution of the two turbulence types varies in function of the altitude and distance to the first row.

The mean kinetic energy $E_k = \frac{1}{2}(\overline{u}^2 + \overline{v}^2 + \overline{w}^2)$, friction velocity $u^* = \sqrt[4]{\overline{u'w'}^2 + \overline{v'w'}^2}$ and turbulent kinetic energy $TKE = \frac{1}{2}(\overline{u'^2} + \overline{v'^2} + \overline{w'^2})$ are estimated at the T tower and indicated in Table 2. All the variables are in good agreement with the experiment at $z = 4m$ (about two times the container height). The friction velocity at $z(T) = 4m$ is at least two times greater than $u^*_{exp}(S) = 0.68 m.s^{-1}$ observed at the S tower feet. That increase is the signature of the turbulence developed by the urban-like canopy. Looking after the results at $z(T) = 8m$ and $z(T) = 16m$, the higher the altitude is the more discrepancies appear between the experimental and numerical results. The experimental measurement of the friction velocity at $z = 16m$ is closed to the upstream value.

| | $E_k (m^2.s^2)$ | | $TKE (m^2.s^2)$ | | $u^* (m.s^{-1})$ | |
|---|---|---|---|---|---|---|
| | MNH-IBM | Yee and Biltoft (2004) | MNH-IBM | Yee and Biltoft (2004) | MNH-IBM | Yee and Biltoft (2004) |
| z=04m | 15.3 | 14.1 | 3.33 | 3.78 | 1.14 | 1.08 |
| z=08m | 36.7 | 29.7 | 1.70 | 3.28 | 0.68 | 0.83 |
| z=16m | 65.8 | 48.5 | 0.02 | 1.75 | 0.02 | 0.60 |

**Table 2.** *Kinetic energy, turbulent kinetic energy and friction velocity obtained by the experimental (Yee and Biltoft, 2004) and numerical (MNH-IBM) investigations at three altitudes of the tower T.*

The Discrete Fourier Transform of the $u$ temporal evolution (Fig. 23) at $z(T) = (4; 8)m$ shows the coherence between the energetic cascade of the experimental investigations and that of the numerical ones. At $z(T) = 16m$, MNH-IBM underestimates the unsteady part of the solution for all wave numbers. The same behaviour is observed on $v$ and $w$ (not shown here).

The MNH-IBM results are consistent with the experimental observations below $z(T) \lesssim 10m$. This well-modelling induces that the turbulence is mostly due to the container wakes upstream the T tower and not to the incoming turbulence (not modelled in the simulation). The influence of the atmospheric turbulence grows with the altitude and MNH-IBM diverges with the experiment. This divergence for $z(T) \gtrsim 10m$ leads to think that the thickness of containers influence is about 4-5 times the height of the urban-like canopy. This MUST case is the subject of ongoing works in our team (Rea et al., In prep).



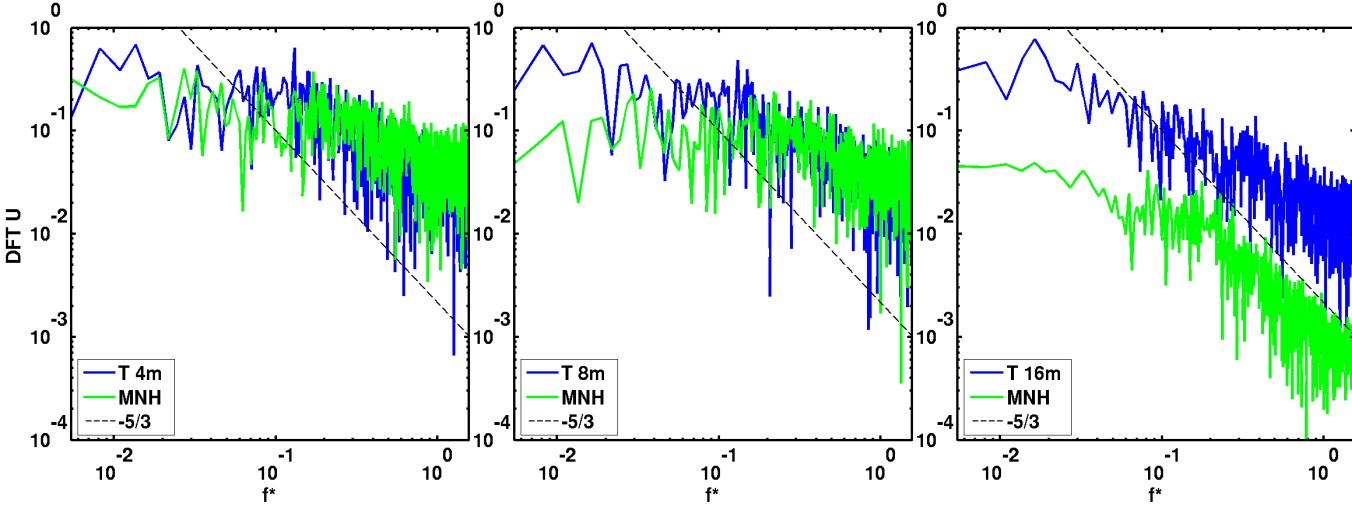

**Figure 23.** *Spectrum of the measured (blue line) and modelled (green line) $u$ wind component at the T tower at $z = (4; 8; 16)m$. $f^*$ is dimensionless as a Strouhal number using $|\mathbf{u}|_{inlet}(z = 2.5m)$ and $z = 2.5m$ the container length.*

## 8 Conclusions and perspectives

This study details the first implementation of an immersed boundary method (IBM) in the atmospheric Meso-NH (MNH) model currently based on mathematical formulations written for structured grids. The MNH-IBM aim is to explicitly model the fluid-solid interaction in the surface boundary layer developed over grounds presenting complex topographies such as cities or industrial sites.

    A LevelSet function (Sussman et al., 1994) characterizes the geometric properties of the fluid-solid interface. Two original

approaches of the GCT Ghost-Cell (Tseng and Ferziger, 2003) and CCT Cut-Cell Techniques (Udaykumar and Shyy, 1995) are implemented to correct the MNH numerical schemes. A newly proposed GCT recovers the fluid information in several image points presenting a distance to the interface independent of the ghost's distance. The CCT consists of a new finite volume approach of fluxes balance near the immersed interface. The GCT is applied to the numerical schemes based on explicit time-integration and the CCT is employed in the implicit resolution of the Poisson equation satisfying the incompressibility

hypothesis. The adaptation and use of iterative procedures solve the pressure problem without any modification to the inverted matrix. The turbulence problem is closed at the fluid-solid interface by a pragmatic LES-RANS formulation based on the subgrid turbulent kinetic energy and the length of the smallest energetic eddies.

    The pressure solver, adapted to the IBM and isolated from the rest of MNH, is used to model potential flows around several obstacles. Compared to analytical and theoretical solutions, the numerical results demonstrate the ability of the IBM adaptation

to ensure that the momentum is preserved and the continuity equation is respected. Non-dissipative flows are simulated to test the IBM forcing of the wind advection scheme (the impact of interpolations collected in Franke (1982), classical and



original GCT comparison and numerical diffusion near interface). These tests validate the original GCT 'three images/ghost points', the use of an inverse distance weighting interpolation near the interface and a trilinear interpolation far from the interface and the modelling of the advection term near the fluid-solid interface by a 2nd-order centered scheme associated with an artificial viscosity calibrated after comparisons with a 3rd-order WENO scheme. With these numerical choices, MNH-IBM demonstrates its ability to model wake instabilities past a circular cylinder placed in a viscous fluid. Then Large-Eddy-Simulations of turbulent flows around bodies with sharp edges and corners are executed (i.e. a cube placed in a channel (Martinuzzi and Tropea, 1993) and a near-neutral atmospheric application over an array of containers (Biltoft, 2001)). These

5   two LES validate the proposed immersed wall-model, switching the characteristic space scale defining the turbulent Reynolds number between one obtained by either a viscous length scale (the cube case) or a roughness length (the containers case).

*Future works*. This study constitutes a first robust step towards a better understanding of the interactions between 'Weather and Cities' and better predictions of such interactions. The idealized character of the physical cases approached here offers some insights. One improvement would consist of a generalization of the IBM writing to terrain-following coordinates (Gal-

10  Chen and Somerville, 1975) allowing for the simulation of high-curvatures bodies in the presence of non-flat grounds. In the run-up of the resolution of multi-scales problems, the consistency with grid nesting (Stein et al., 2000) would be pertinent as well as the coupling to a drag model (Aumond et al., 2013). In the current paper, 'simple' bodies are investigated; the modelling of real houses and buildings with arbitrary shape in close proximity to each other is ongoing with works dedicated to a brief and intense pollutant episode due to a factory explosion in 2001 over Toulouse (Auguste et al., in prep.) assuming a dry neutral case with a non-reactive gas dispersion. Such hypotheses involve a broad range of physical phenomena requiring numerical compliances with IBM including the chemical reactions, phase changes and radiative effects. Such compliances would allow

the access to a large variety of atmospheric situations.

## 9   Code availability

The immersed boundary method has been implemented in the 5.2 version of the Meso-NH code. This reference version is under the CeCILL-C license agreement and freely available at http://mesonh.aero.obs-mip.fr/mesonh52. The source files dedicated to IBM are currently accessible on a CERFACS server (contact the corresponding author). It will be integrated in a future

Meso-NH version.

*Acknowledgements.*   We thank for stimulating and fruitful discussions: J. Magnaudet (Institut de Mécanique des Fluides, IMFT, Toulouse), Y. Hallez (Laboratoire de Génie Chimique de Toulouse, LGC, Toulouse), J.-L. Pierson (Institut Francais Pétrolier 'Energies Nouvelles', IFPEN, Lyon), J.-L. Redelsperger (Laboratoire de Physique des Océans, IFREMER, Brest), J.-P. Pinty and J. Escobar (Laboratoire d'Aérologie, LA, Toulouse), O. Thouron, T. Lunet, L. Giraud, M. Rochoux, O. Vermorel, I. d'Ast and G. Dejean (Centre Européen de Recherche et Formation

Avancée en Calcul Scientifique, CERFACS, Toulouse). The simulations were performed on the Neptune/Nemo-CERFACS supercomputers, on the Occigen-CINES Bull cluster (c2016017724 and a0010110079 GENCI projects). A part of this work was supported by the 'Région Midi-Pyrénées' funding.





## Appendix A:  Interface basis change

Velocity vector $\overline{\mathbf{u}}$ known in the Cartesian mesh basis $(\mathbf{e_x}, \mathbf{e_y}, \mathbf{e_z})$ at mirror $I$ and at images $I_l$ ($\Delta n_1 = \Delta$ and $\Delta n_2 = 2\Delta$ in

Fig. 2-a) have to be projected in the basis of the interface $(\mathbf{e_n}(B), \mathbf{e_t}(B), \mathbf{e_c}(B))$ in which boundary conditions on each vector component are imposed. $B$ is the interface point associated to the ghost point $G$ as it illustrates in the figure. Computing the gradient of the LevelSet function, the normal direction is known $\mathbf{e_n}(B) = \mathbf{e_n}(G)$ and $\mathbf{e_n} = -\overrightarrow{\nabla}\phi / \| \overrightarrow{\nabla}\phi \|$. $(\mathbf{e_t}, \mathbf{e_c})$ are currently two arbitrary tangent directions. To fix the tangent directions, $\mathbf{e_t}$ is considered as the predominant tangent direction of the flow along the fluid-solid interface depending on the direction observed at the studied mirror/images by defining the

velocity vector as $\overline{\mathbf{u}}(I_l) = \overline{u}_n(I_l)\mathbf{e_n} + \overline{u}_t(I_l)\mathbf{e_t}(I_l)$. This writing induces:

$$\mathbf{e_c}(I_l) = \frac{\mathbf{e_n} \otimes \overline{\mathbf{u}}(I_l)}{\| \mathbf{e_n} \otimes \overline{\mathbf{u}}(I_l) \|} \tag{A1}$$

$$\mathbf{e_t}(I_l) = \mathbf{e_c}(I_l) \otimes \mathbf{e_n} \tag{A2}$$

The $(\mathbf{e_n}, \mathbf{e_t}, \mathbf{e_c})$ basis at the interface could be defined by considering that the velocity vector does not rotate around $\mathbf{e_n}$

between the first node and the interface (constant) or by considering $\mathbf{e_t}(B)$ as a linear interpolation of those obtained at the $I_l$ image points (linear):

$$\mathbf{e_t}(B) = \mathbf{e_t}(I_1)(constant); \mathbf{e_t}(B) = 2\mathbf{e_t}(I_1) - \mathbf{e_t}(I_2)(linear) \tag{A3}$$

$$\mathbf{e_c}(B) = \mathbf{e_n} \otimes \mathbf{e_t}(B) \tag{A4}$$

Finally the velocity vector $\overline{\mathbf{u}}$ at mirror and/or images is projected on the specified $(\mathbf{e_n}(B), \mathbf{e_t}(B), \mathbf{e_c}(B))$ basis. The desired component of $\overline{\mathbf{u}}(G)$ is calculated firstly in the interface frame and secondly by an inverse projection in the referent mesh frame.

## Appendix B:  3D interpolations

Four interpolation techniques are implemented to recover the desired variable $\psi(I_l)$ at the images and/or mirror:

- The Polynomial Lagrange Interpolation (PLI)
- The Barycentric Lagrange interpolation (BLI)

- The Inverse Distance Weighting Interpolation (IDW)
- The Modified Distance Weighting Interpolation (MDW)

$\psi(x_l)$ is the $\psi$-value at the image point of $x_l$ location. $\psi(x_i)$ refers to the known values localized at the mesh nodes.

**Polynomial Lagrange Interpolation.** Let $N$ the number of nodes ($i = [1, N]$) and $L_i$ the Lagrange polynomial of the node $x_i$ defined for the $x_l$ image/mirror, the formulation of the Lagrange interpolation is:



$$L_i(x_l) = \prod_{p=1, p \neq i}^{N} \frac{x_l - x_p}{x_i - x_p} ; \psi(x_l) = \sum_{i=1}^{N} L_i(x_l).\psi(x_i) \qquad \text{(B1)}$$

The extension to 3D case ($\mathbf{x_l} = [x_l, y_l, z_l]$) is called the trilinear interpolation ($PLI$) and consists in three successive steps:

$$L_i(x_l) = \prod_{p=1, p \neq i}^{N} \frac{x_l - x_p}{x_i - x_p} ; L_j(y_l) = \prod_{p=1, p \neq j}^{N} \frac{y_l - y_p}{y_j - y_p} ; L_k(z_l) = \prod_{p=1, p \neq k}^{N} \frac{z_l - z_p}{z_k - z_p} \qquad \text{(B2)}$$

$$\psi(x_l, y_l, z_l) = \sum_{i=1}^{N} \sum_{j=1}^{N} \sum_{k=1}^{N} L_i(x_l) L_j(y_l) L_k(z_l).\psi(x_i, y_j, z_k) \qquad \text{(B3)}$$

**Barycentric Lagrange interpolation.** The Lagrange polynomials ($BLI$) are re-written in the 1D case (the 3D extension is direct):

$$L^B(x_l) = \left( \sum_{i=1}^{N} \frac{1}{(x_l - x_i) \prod\limits_{p=1, p \neq i}^{N} (x_i - x_p)} \right)^{-1} ; \psi(x_l) = L^B(x_l) \sum_{i=1}^{N} \frac{\psi(x_i)}{(x_l - x_i) \prod\limits_{p=1, p \neq i}^{N} (x_i - x_p)} \qquad \text{(B4)}$$

**Inverse Distance Weighting Interpolation.** $\psi(x_i)$ are predominantly weighted by a $x_l$ distance function to its closest

neighbours $\psi(x_l \approx x_i)$. An $\alpha$ exponent parameter is operating to modulate the vicinity ($IDW$):

$$\psi(x_l) = \frac{\sum\limits_{i=1}^{N} L_i^D(x_l).\psi(x_i)}{\sum\limits_{i=1}^{N} L_i^D(x_l)} ; \mid \mathbf{x_l} - \mathbf{x_i} \mid = \sqrt{(x_l - x_i)^2 + (y_l - y_i)^2 + (z_l - z_i)^2} \qquad \text{(B5)}$$

where $L_i^D(x) = \frac{1}{|x_l - x_i|^\alpha}$. This formulation diverges when $x_i \to x_l$ and it is commonly adopted to impose $\psi(x_l) = \psi(x_i)$ when $\exists (x_i - x_l) \leq \epsilon$ ($\epsilon$ is an arbitrary parameter depending on the mesh discretization, $\epsilon << \Delta$). The 3D extension is direct.

**Modified Distance Weighting Interpolation.** The modified version of the $IDW$ noted ($MDW$) is referred in Franke (1982) and defines a $\beta = 2\Delta$ length (or radius in 3D) taking the hypothesis that $\psi(x_l - x_i > \beta)$ does not affect $\psi(x_l)$ (ie $L_i^M(\mid x_l - x_i \mid > \beta) = 0$):

$$\psi(x_l) = \frac{\sum\limits_{i=1}^{N} L_i^M(x_l).\psi(x_i)}{\sum\limits_{i=1}^{N} L_i^M(x_l)} ; L_i^M(x_l) = (\frac{\beta - \mid x_l - x_i \mid}{\beta \mid x_l - x_i \mid})^\alpha \qquad \text{(B6)}$$

**Appendix C: 1D interpolation**

1D interpolation is claimed in the direction normal to the interface to estimate the $\psi(G)$ value. The classical mirror point is $I$ of $\psi(I)$ value (Fig. 2) and LSF $\phi(G) = -\phi(I) = \phi$. The interface point is $B$ ($\phi(B) = 0$). The desired $\psi(B)$ (resp. $\frac{d\psi}{dn}\big|_B$) interface





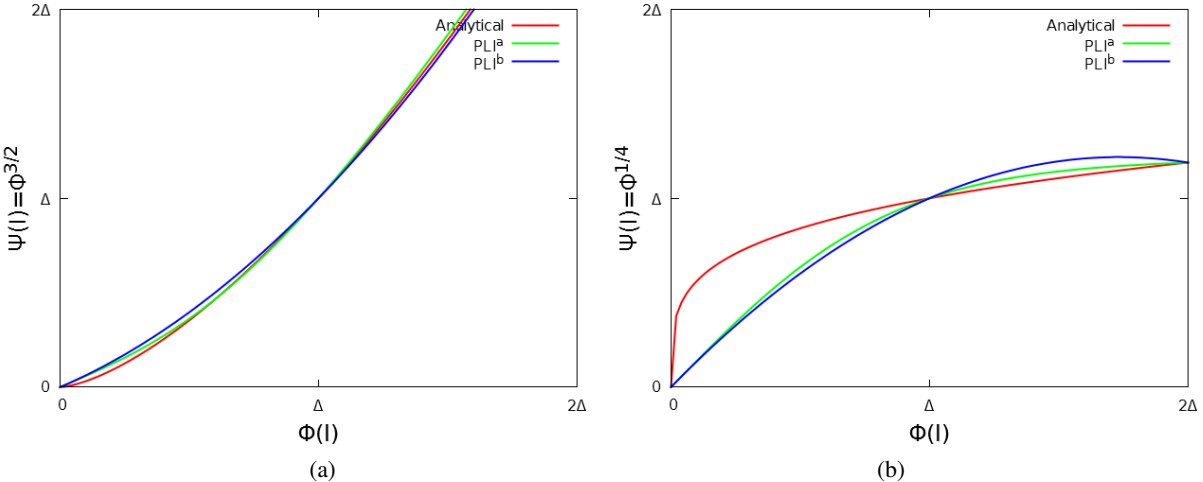

**Figure 24.** *Quadratic interpolations of two analytical profiles (red lines) using two images nodes $\phi = -[1;2]\Delta$ and the interface node: Green (resp. blue) using $L^a$ (resp. $L^b$) polynomials.*

condition is known for a Dirichlet (resp. Neumann) condition and $\psi(G) = 2\psi(B) - \psi(I)$ (resp. $\psi(G) = 2\phi\frac{d\psi}{dn}\big|_B + \psi(I)$). The mirror technique estimates $\psi(I)$ by a 3D interpolation (Appendix B). Another way to recover $\psi(I)$ consists in a quadratic reconstruction $f()$ using two image points $(I_1, I_2)$ of $(\psi(I_1), \psi(I_2))$ values. Two distinct calculations of $f(\psi(B), \psi(I_1), \psi(I_2))$ noted $PLI^a$ and $PLI^b$ are used to build a 1D Lagrange interpolation:

$$\psi^a(I) = (2L_G^a(I)\psi(B) + L_{I_1}^a(I)\psi(I_1) + L_{I_2}^a(I)\psi(I_2))\frac{1}{1 + L_G^a(I)} \tag{C1}$$

$$\psi^b(I) = L_B^b(I)\psi(B) + L_{I_1}^b(I)\psi(I_1) + L_{I_2}^b(I)\psi(I_2) \tag{C2}$$

where $L^a(I)$ (resp. $L^b(I)$) are the Lagrange polynomials:

$$L_{I_1}^a(I) = (\frac{2\Delta - \phi}{\Delta})(\frac{2\phi}{\Delta + \phi}); L_{I_2}^a(I) = (\frac{\phi - \Delta}{\Delta})(\frac{2\phi}{2\Delta + \phi}); L_G^a(I) = (\frac{\phi - \Delta}{\phi + \Delta})(\frac{\phi - 2\Delta}{\phi + 2\Delta}) \tag{C3}$$

$$L_{I_1}^b(I) = (\frac{2\Delta - \phi}{\Delta})(\frac{\phi}{\Delta}); L_{I_2}^b(I) = (\frac{\phi - \Delta}{\Delta})(\frac{\phi}{2\Delta}); L_B^b(I) = (\frac{\phi}{\Delta})(\frac{\phi - 2\Delta}{2\Delta}) \tag{C4}$$

The behaviour of each quadratic interpolation and their abilities to approach a power law such $\psi = \phi^{3/2}$ (resp. $\psi = \phi^{1/4}$) is illustrated in Figure 24-a (resp. Fig. 24-b). $PLI^a$ fits better the analytical solution in the two cases and is adopted in the rest of this paper.



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
