# Peer review of "Implementation of an Immersed Boundary Method in the Meso-NH model: Applications to an idealized urban-like environment."

_Geoscientific Model Development, 2018_

## Short Comment (SC1) · 20 Mar 2018

Dear authors,

in my role as Executive editor of GMD, I would like to bring to your attention our Editorial version 1.1: http://www.geosci-model-dev.net/8/3487/2015/gmd-8-3487-2015.html This highlights some requirements of papers published in GMD, which is also available on the GMD website in the 'Manuscript Types' section: http://www.geoscientific-model-development.net/submission/manuscript_types.html

In particular, please note that for your paper, the following requirement has not been met in the Discussions paper:

- "The main paper must give the model name and version number (or other unique identifier) in the title."

Please provide the version number of IBM and of the MESO-NH model in the title of your revised manuscript.

As explained in https://www.geoscientific-model-development.net/about/manuscript_types.html GMD is encouraging that authors to upload the program code of models (including relevant data sets) as supplement or make the code and data of the exact model version described in the paper accessible through a DOI (digital object identifier). In case your institution does not provide the possibility to make electronic data accessible through a DOI you may consider other providers (eg. zenodo.org of CERN) to create a DOI. Please note that in the code accessibility section you can still point the reader how to obtain the newest version.

If for some reason the code and/or data cannot be made available in this form (e.g. only via e-mail contact) the "Code Availability" section need to clearly state the reasons for why access is restricted (e.g. licensing reasons).

Yours, Astrid Kerkweg

---

## Referee Comment (RC1) · Anonymous Referee #1 · 29 May 2018

[10pt,a4paper]article [utf8]inputenc amsmath amsfonts amssymb [lmargin=2.5cm, rmargin=2.5cm]geometry

**Comments : Auguste et al. 2018**

May 29, 2018

**1   General comments**

This manuscript presents an Immersed Boundary Method implemented in the Meso-NH model. The authors first focus on the implementation of the boundary technique. Following this, the authors turn to a detailed validation of the code.

I would like to compliment the authors for presenting such an extensive work. Implement a ghost-cell / cut-cell technique is not an easy task and the authors are doing good. Moreover the implementation of IB techniques in forecast weather code seems to be very promising according to the obtained results.

That's said, some additional things absolutely need to be added/modified to make this presentation clearer from a numerical point of view. As it will be mention in the following paragraphs some variables are not defined distinctly. Numerical details are lacking (test of Level Set function, needs of a second mirror points in the ghost cell technique, ...). When reading the numerical part the reader get frustrated from this lack of details. Moreover, despite it's interest for fluid mechanics purpose, and especially for validating the mass conservation, the potential flow around a cylinder does not seems to be

adapted to the present journal. For reasons which will be explain in the paragraphs below I recommend the author to rewrite the numerical parts and the test cases 4,5,6 with much more details in a supplementary materials or in the journal dedicated to numerical fluid mechanics. The newly paper submitted to GMD should contain a short presentation of the numerical method with a reference to the supplementary materials (or paper) and the cases of section 7.

**2 Specific comments**

**2.0.1 Abstract**

Nothing to report

**2.0.2 1 Introduction**

Excepts the minor few points reported below the introduction is well written.

- *This study describes the numerical implementation, verification and validation of an Immersed Boundary Method (IBM) in the atmospheric solver Meso-NH for applications to urban flow modelling.* needs some reference

- *Covering all possible cases is obviously impossible but from a fluid mechanics standpoint one can invoke the principle of similarity which permits, for example, to observe von Kármán streets in the wake of a centimeter-scale cylinder as well as in the cloud layout behind an island.* I understand the thought of the authors, but in both cases the Reynolds number (which characterize the "principle of similarity") is much different.

[Figure]

- *Even if the physical application in our mind is the atmospheric mesoscale reaction to perturbations induced by urban cities, the more the obstacles are considered as a part of the scales numerically resolved the more the results accuracy is.* Is this sentence is useful ?

- *This approach and its variant developed by Goldstein et al. (1993) for a rigid interface can suffer from the lack of stiffness (fluid-solid interface is generally spread over few cells) which can be problematic to recover 5 the boundary layer.* The main issue with Goldstein et al. (1983) approach comes from the fact that the time step should be very low if we want to enforce a rigid boundary condition.

2.0.3   5 Potential flows

- *Depending on how to resolve the partial differential equations, Cartesian grid methods (Ye et al., 1999) are written for finite-volume discretizations (Cut-Cell Technique, CCT) and for finite-difference discretizations (Ghost-Cell Technique, GCT) as in Tseng and Ferziger (2003).* This sentence could be clarified.

- *Another argument in favor of discrete forcing is that it does not introduce source terms in the conservation equations.* Source terms are introduced but in a discrete way rather than in a continuous way.

2.0.4   2 The Meso-NH code at a glance

The overall section is understandable but some definitions are missing : $\mathbf{u}$, $s$,... The additional term such as $\mathbf{f}_\theta$ should be explicitly given or not mentioned. Indeed for a significant part of the paper (potential flow, DNS, ...) Coriolis forces, molecular diffusion does not play any role. Moreover more details about the LES technique could be given, for instance what is the filter size ?

Finally, it is interesting to know that the advective term can be solved using 4 different schemes to understand section 5. But the paper will gain in clarity if those details were mentioned in another paper / report.

2.0.5 3 The IBM forcing in the Meso-NH code

This part is the less understandable. We distinguishably see the large amount of work done by the authors to implement the methods but too few details are given and some paragraph are misunderstanding.

- *An intensive study had been done (not shown here) to verify the ability of the LSF spatial derivatives to recover the vector normal to the interface* this is really interesting and those results should be given. Indeed other authors (Kempe & Froelich JCP 2012) are using LS method to follow the shape of moving immersed bodies.

- *l3-l8* this paragraph should be clarified. Is it necessary to mention the Heaviside function as function of the LS function ? The authors could say that there are only looking at point closed to the interface.

- *A problematic case regularly met in the mirror interpolation is the vicinity of ghosts with the interface.* Why ?

- *The definition of several images per ghost permits the access to a building of the normal profile of the fluid information by an 1D quadratic interpolation* This point could be clarified.

- *Note the $\frac{l\phi}{2}$ -approximation on the location of the derivative term.*This point could be clarified.

- *First looking at the RHS of (5), the ... law coming from the resolution of the explicit-in-time schemes near the interface and in the solid regions badly affects the ... computation (note that the GCT operates after the step projection).* To be clarified.

- the meaning of the symbol $\widetilde{u}$ could be clarified

- Why the disk radius is split in 8. Why not 4 or 16 ? Figure 4 (b) is misunderstanding since the arbitrary surface is a square !

- *This number is limited by a convergence criterion (compromise between incompressibility satisfaction and CPU cost).* What is the CPU cost of this step ?

- The subsection 3.3 could be clarified. The referee is not fully accustomed to wall turbulence modelling. Nevertheless the discussion is quite hard to follow.

Since the proposed methods are closed to the ones proposed by Kim, Kim & Choi (JCP 2001) the authors could mention it in the introduction.

2.0.6   4 Potential flows

I found this section really interesting. I just mention a few minor points to clarified :

- *With a change of Galilean reference frame, this study corresponds to an uniform body acceleration ab in a fluid initially at rest* The steady uniform flow past a sphere is not equivalent to the accelerated flow past a sphere. How the added mass is computed ?

- *Unsurprisingly Acyl increases with the confinement.* Reference ?

- Taylor-Green vortices are solutions of the full 2D Navier-Stokes equations. I don't understand their location in the potential flow section.

- *To conclude, this section validates the modification to the pressure solver. IBM appears less accurate than BFM when the ground presents low curvature in regard of the space resolution. It seems more pertinent than BFM to model high interfaces such as sharp edges or corners.* More details should be given to explain this behaviour.

**2.0.7   5 Potential flows**

- *The adopted strategy with IBM is to model the advection term with a low order scheme near the interface (Sect. 3).* To avoid numerical diffusion, why a low order scheme is used ?

- *The vorticity equation for a 2D inviscid flow reveals no production in time.* To be clarified.

- *To fit as well the potential solution, a non-trivial condition is employed on the tangent velocity ...* Why such a complicated boundary condition is used ?

**2.0.8   6 DNS**

Nothing to report except give more details on how the viscous term added in momentum equation is computed.

**2.0.9   7 8 9**

Nothing to report.

---

## Referee Comment (RC2) · Anonymous Referee #2 · 7 Oct 2018

General comments

The authors have implemented an immersed boundary method for modeling complex obstacles in the atmospheric code Meso-NH. The proposed method consists of a modified ghost-cell technique in the velocity prediction step and a cut-cell technique in the projection step. Finally a wall model is presented along with the large eddy simulation. The proposed method is then validated through a wide range of benchmark tests.

In sum, the work is quite extensive and involves a lot of numerical techniques. But the implementation details are not clearly expressed. The illustrative figures are not well presented. Some of the notations used in the paper are also confused. This article can

be improved by taking the following suggestions into consideration.

Specific comments

Section 3

In Figure 1 (a), what does the triangle mean?

The authors should give the mathematical expression of the level set function and show how it is used to identify the interface by one example, as did in the cited paper of Tseng and Ferziger (2013).

Section 3.1

The authors have suggested to use more image point to avoid numerical issues when the ghost node is close to the boundary. The idea is new but not clearly demonstrated. Why this becomes necessary and how the new idea can solve the numerical problem ?

In the expression GI = 2 phi_G n, the variable "phi_G" is not defined.

The Figure 2 is not well illustrated, as the reader could not easily identify the difference of different approaches.

The use of "original" for the proposed method to differ from the classical method would be misleading and inappropriate. The author could use another word like "new", "novel" or "proposed" instead.

Section 4

The Taylor vortex or decaying vortex is considered as a potential flow as an accuracy test. However it usually refers to an unsteady rotational flow and hence not a potential flow.

Section 6

"A molecular diffusion is explicitly added as a source term in Eq. (3) to achieve a

converged in time and in space solution of a non-linear problem." This expression is confused as the molecular diffusion is just added for numerical purpose instead of from the physical viewpoint.

In the flow over a stationary cylinder test, the authors have studied different meshes but not indicated the domain size, while which has much greater impact to the final solution. Comparison could be made to the reference paper "Moving immersed boundary method, International Journal for Numerical Methods in Fluids, 2017".

A convergence study could be performed for the convergence rate of velocity, as it never shows in the other parts.

---

## Author Comment (AC1) · 3 Nov 2018

**Introduction**

We thank the Referee 1 for his/her interest in our work and his/her positive appreciation of the manuscript. We are glad that the Referee 1 gives some suggestions to improve the manuscript. Our response is split in two sections. The first section answers to the similar comments done both by the Referee 1 and by the Referee 2. The second section gives the responses related to the specific comments done by the Referee 1. As the discussion progresses, we invite the Referee 1 to look after the revised version

of the manuscript sent with the present document. A color code applied to the text highlights the modifications: the red color is used when modifications/corrections are done; the green color is used when new insertions are proposed. Note the modification of the title following the GMD requirement.

**1   Common response to Referee 1 and Referee 2**

*The Referees compliment the extensive work but feel that the details of the numerical implementation are not clearly expressed. Moreover, the numerical implementation suffers from a lack of details.*

To propose an immersed boundary method (IBM) in the Meso-NH (MNH) code able to model the ground or topography interaction with an atmospheric flow, as it mentions by the Referees, the number of numerical developments and associated validations has to be high. It induces a long description which could be problematic regarding the format of a manuscript. For this reason, the authors decided to condense the ample information running the risk of loosing a part. The authors agree with the Referees observations. Therefore, in the proposed revised version, an important effort to give additional details is done. About the organization, Referee 1 suggests either to split the paper in two parts and/or to place the current Sections 4, 5 and 6 (which are pointed out in the limit of the GMD scope) in Appendix. Referee 2 requires more details on the cylinder case at moderate Reynolds number (Sect. 6). The authors share the same view and propose to preserve one test case dedicated to the forcing of the pressure solver and to place the others test cases of Sections 4 and 5 in Appendix. Following the Referee 2 and because of the GCT validation, Section 6 is preserved in the core of the paper. The new structure of the paper allows to detail the numerical methods (Sect. 2 and 3) without an increase of the paper volume.

In particular, the Referees make it clear the lack of details on the use of the image points in the Ghost-Cell Technique (GCT) and on the Level-Set Function (LSF). The discussion on GCT in Section 3 is therefore reinforced. Concerning LSF and in the present paper, this function is built for academic bodies and their theoretical solutions are known. The intensive work we had done to implement an accurate LSF was related to the modeling of an interface only known in a discrete way (such as the data of a real urban topography). This work is presented in another paper: Auguste et al.(submitted to Atmospheric Environment). That's said, the LSF presentation in the present paper is reinforced in Section 3.

*The Referees mention that the simulation of the Taylor-Green vortex does not have to appear in the section dedicated to the potential flows.*

The Referees are absolutely right. At the short time scales, the viscous influence vanishes and the Taylor-Green vortex solution is associated to an inviscid flow. The confusion (and mistakes) of the authors to put this test case in a "Potential Flows" category comes from an abusive use of the "Taylor-Green vortex" term. Even if the flow structure solution presents an array of vortex similar to the Taylor-Green ones, we do not have the right to use this term. The studied potential flow (the velocity field derives from $\pi^{-1}cos(\pi lx)sin(\pi my)$) is solution of the Poisson equation (Popinet, 2003). This case testing the pressure solver moves into Appendix of the proposed revised manuscript.

*The Referees mention a lack of details and/or confusion on the molecular diffusion used in the Direct Numerical Simulations.*

The molecular diffusion is taken into account in the cases presenting a low Reynolds number ("low" compared to most of $Re$ of atmospheric applications). This term is associated to the fluid kinematic viscosity $\nu_f$. Therefore, $\nu_f \Delta u$ is explicitly added

in a physical purpose (Navier-Stokes equations resolution, Eq. 3). The numerical implementation is the most simple: in 1D for example, its contribution on the $\frac{\partial u_i^{n+1}}{\partial t}$ is computed for uniform Cartesian grids such as $\nu_f/\Delta^2(u_{i+1}^n - 2u_i^n + u_{i-1}^n)$ where $\Delta$ is the space step. The explicit-in-time resolution induces the respect of the stability condition $\mathcal{O}(\nu_f/\Delta^2)$. Some additional comments about the fluid viscosity are inserted in Section 2. Even if this type of flow is far from the atmospheric applications, this is a robust way to test and validate the implemented GCT. For example, this study makes us confident in the forcing of the Reynolds stresses $\nabla.(\nu_e \nabla \overline{u})$ near an immersed wall ($\nu_e$ the turbulent viscosity). In the same spirit and for future thermodynamic applications, the authors mention that another study was carried out on the parabolic heat equation to confront (and validate) the GCT to a 1D pure diffusion problem $\frac{\partial T_f}{\partial t} = (\lambda_f/\rho C_p)\Delta T_f$.

**2 Specific response to Referee 1**

*"This study describes the numerical implementation, verification and validation of an Immersed Boundary Method (IBM) in the atmospheric solver Meso-NH for applications to urban flow modelling". Need some reference.*

The IBM annual review of Mittal and Iaccarino (2005) and the MNH reference papers (Lafore et al., 1998; Lac et al. , 2018) are inserted.

*"Covering all possible cases is obviously impossible but from a fluid mechanics standpoint one can invoke the principle of similarity which permits, for example, to observe von Kármán streets in the wake of a centimeter-scale cylinder as well as in the cloud layout behind an island." I understand the thought of the authors, but in both cases the Reynolds number (which characterize the "principle of similarity") is much different.*

[Figure]

As it is suggested by the Referee 1, the use of the "principle of similarity" is abusive. The sentence is rephrased : "Covering all possible cases is obviously impossible but from a fluid mechanics standpoint one can invoke the persistence of flow behaviors whatever the scales. It allows for example, to observe von Kármán streets in the wake of a centimeter-scale cylinder as well as in the cloud layout behind an island."

*"Even if the physical application in our mind is the atmospheric mesoscale reaction to perturbations induced by urban cities, the more the obstacles are considered as a part of the scales numerically resolved the more the results accuracy is." Is this sentence is useful ?*

The sentence is rephrased: "The physical application in our mind is the atmospheric mesoscale reaction to perturbations induced by urban cities and the more the obstacles are considered as a part of the scales numerically resolved the more the results accuracy is."

*"This approach and its variant developed by Goldstein et al. (1993) for a rigid interface can suffer from the lack of stiffness (fluid-solid interface is generally spread over few cells) which can be problematic to recover the boundary layer." The main issue with Goldstein et al. (1983) approach comes from the fact that the time step should be very low if we want to enforce a rigid boundary condition.*

The sentence is rephrased: "This approach and its variant developed by Goldstein et al. (1993) for a rigid interface can suffer from the lack of stiffness (fluid-solid interface is generally spread over few cells) and the time step restriction."

*"Depending on how to resolve the partial differential equations, Cartesian grid methods (Ye et al., 1999) are written for finite-volume discretizations (Cut-Cell Technique, CCT) and for finite-difference discretizations (Ghost-Cell Technique, GCT) as in Tseng and*

[Figure]

*Ferziger (2003)." This sentence could be clarified.*

A sentence is added: "CCT reshapes the cell cut by the interface to preserve mass, momentum and energy. Using GCT, the local spatial reconstruction is done inside the solid region."

*"Another argument in favor of discrete forcing is that it does not introduce source terms in the conservation equations." Source terms are introduced but in a discrete way rather than in a continuous way.*

We agree with the Referee. The authors discuss about a GCT characteristics and apply incorrectly this to all discrete forcing types. The sentence is rephrased: "An argument in favor of GCT is that it does not introduce source terms in the conservation equations."

*The Meso-NH code at a glance. The overall section is understandable but some definitions are missing and should be explicitly given or not mentioned. Indeed for a significant part of the paper (potential flow, DNS, ...) Coriolis forces, molecular diffusion does not play any role. Moreover more details about the LES technique could be given, for instance what is the filter size ? Finally, it is interesting to know that the advective term can be solved using four different schemes to understand section 5*

The terms which are not used in the present paper disappears in the text and in the equations. To improve the presentation of the MNH turbulence scheme (Cuxart et al., 2000), Section 2.3 is reinforced. To improve the introduction of the IBM forcing of the turbulence scheme, Sect. 3.3 is reinforced.

*Other authors (Kempe and Froelich JCP 2012) are using LS method to follow the shape of moving immersed bodies.*

The reference Kempe and Frölich (2012) is added during the LSF presentation.

*"l3-l8" - This paragraph should be clarified. Is it necessary to mention the Heaviside function as function of the LS function ? The authors could say that there are only looking at point closed to the interface.*

The heaviside function was used to clearly show that the forcing acts only in the solid region and the conservation laws in the fluid region is preserved. This use is dispensable. Equation 7 disappears and Fig. 1 is modified in consequence.

*"A problematic case regularly met in the mirror interpolation is the vicinity of ghosts with the interface". Why ?*

A ghost in the vicinity of the interface has an image point also closed to the interface. This is not a problem when the boundary condition is independent of the fluid information. This could be a problem when wall models (based on the turbulent Reynolds number for example) are used. In the worst case scenario, the ghost and its associate mirror point coincide.

*"The definition of several images per ghost permits the access to a building of the normal profile of the fluid information by an 1D quadratic interpolation". This point could be clarified.*

The authors hope the clarity of Section 3.1 is improved in the revised version.

*"Note the $l\phi/2$ approximation on the location of the derivative term". This point could be clarified.*

The formulation is exact when $\frac{\partial \overline{\psi}}{\partial n}|_b = \frac{\partial \overline{\psi}}{\partial n}|_{l\phi/2}$

*"First looking at the RHS of (5), the ... law coming from the resolution of the explicit-in-time schemes near the interface and in the solid regions badly affects the ... computation (note that the GCT operates after the step projection)." To be clarified.*

If the CCT is not used to correct the pressure solver, the computation of the RHS of Eq. (5) will perceive the GCT. Some old tests had shown that the impermeability satisfaction using GCT was not convincing. An imposition of a null velocity in the solid region was more convincing but it raised the question of the mass and energy conservations in the truncated cells. These observations induced the motivation to correct the pressure solver with a volume approach and a reconstruction of the truncated cells.

*The meaning of the symbol tilde could be clarified.*

The tilde symbol was used to distinguish the tangent velocity at the interface without and with the pragmatic limitation. To avoid some possible confusion, the use of the symbol disappears in the revised paper.

*Why the disk radius is split in 8. Why not 4 or 16 ? Figure 4 (b) is misunderstanding since the arbitrary surface is a square !*

The choice of the splitting was decided regarding the number of adjacent points and desiring an isotropic behavior of the surface integration. The Referee correctly points out that this choice is an arbitrary one and a study of the impact of the splitting type should be perspicacious. The good results of the initial choice do not motivate this sensitivity study in the present paper. This study could occur in the future when this technique will be generalized to non uniform Cartesian grids.

*"This number is limited by a convergence criterion (compromise between incompressibility satisfaction and CPU cost)". What is the CPU cost of this step ?*

With a low number of transported species, the numerical schemes in MNH responsible of the CPU cost are mainly the advection of the mean wind and the pressure solver. Depending on the flow, the Poisson equation in the pressure solver is resolved by $[2:4]$ iterations. The characteristic CPU cost of the pressure solver is about $[20:30]\%$ the global cost.

*The subsection 3.3 could be clarified. The referee is not fully accustomed to wall turbulence modelling. Nevertheless the discussion is quite hard to follow.*

The authors hope the clarity of Section 3.3 is improved in the revised version.

*Since the proposed methods are closed to the ones proposed by Kim, Kim and Choi (JCP2001) the authors could mention it in the introduction.*

The reference Kim et al. (2001) is added.

*Potential flows. "With a change of Galilean reference frame, this study corresponds to an uniform body acceleration ab in a fluid initially at rest The steady uniform flow past a sphere is not equivalent to the accelerated flow past a sphere". How the added mass is computed ?*

The potential scalar $\Psi^*$ evolves in the solid region and induces a modification of the velocity field. For an acceleration during one time step (the body translating in a longitudinal direction), $\Psi^*$ corresponds to a longidutinal gradient solution and $\mathcal{A}m_f a_b = \int_{\mathcal{V}_s} \frac{\partial \rho_f u}{\partial t} d\mathcal{V}$; the equation is completed (Sect. 4.1).

*"Unsurprisingly Acyl increases with the confinement". Reference ?*

The reference Brennen (1982) is added.

[Figure]

*"To conclude, this section validates the modification to the pressure solver. IBM appears less accurate than BFM when the ground presents low curvature in regard of the space resolution. It seems more pertinent than BFM to model high interfaces such as sharp edges or corners". More details should be given to explain this behaviour.*

The sentence is inserted: "The minimum pressure which should aim to $-\infty$ is not perceived or smoothed by IBM and allows the pressure solver not to diverge. Note that Lundquist et al. (2010, 2012) using compressible WRF model observe similar behaviors." This sentence appears now in Appendix related to the pressure solver test.

*"The adopted strategy with IBM is to model the advection term with a low order scheme near the interface (Sect. 3). To avoid numerical diffusion, why a low order scheme is used ?"*

The order in space of the numerical scheme dedicated to the mean wind is decreased near the immersed interface such as the scheme order is decreased in BFM. The decrease in IBM is not essential but allows to limit the number of ghosts in the solid region, limit the communications during a parallel computation. Indeed, the chosen implementation implies that the associated images and ghosts points have to be localized in the integration volume of each processor. The cells thickness to communicate between processors is currently four. A part of this comment is added in the Appendix A.

*"The vorticity equation for a 2D inviscid flow reveals no production in time". To be clarified.*

The vorticity equation ($w$, the vorticity) in the studied case (inviscid flow, uniform density, incompressible) is $\frac{\partial w}{\partial t} + (\overline{u}.\nabla)w = -(w.\nabla)\overline{u}$. The LHS term vanishes in a 2D case.

*"To fit as well the potential solution, a non-trivial condition is employed on the tangent velocity" Why such a complicated boundary condition is used.*

The velocity field $\overline{u}$ evolves such as $\mathcal{O}(1/r^2)$ with $r$ the radial distance to the cylinder interface. To not impose a linear evolution of $\overline{u}$ and $\frac{\partial \overline{u}}{\partial r}$, we use a simple wall model acting such as a weak condition. This case is also a good example of the interest to define several images points.

**References**

Auguste, F. (2010). Instabilités de sillage et de trajectoire dans un fluide visqueux. Ph.D. thesis, University of Toulouse.

Auguste, F., Paoli, R., Lac, C., Masson, V., and Cariolle, D. (submitted to Atmospheric Environment). Large eddy simulations devoted to the health impact of pollutant dispersions in cities: the case of the NO2 plume due to the AZF explosion in Toulouse (21/09/01).

Brennen, C. E. (1982). A Review of Added Mass and Fluid Inertial Forces. Naval Civil Engineering Laboratory, Port Hueneme, CA. CR 82.010.

Cai, S.-G., Ouahsine, A., Favier, J., and Hoarau, Y. (2017) Moving immersed boundary method. Int. J. Numer. Methods Fluids, 85(5), 288-323.

Cuxart, J., Bougeault, P., and Redelsperger, J.-L. (2000). A turbulence scheme allowing for mesoscale and large-eddy simulations. Quart. J. Roy. Meteor. Soc., 126(562), 1-30.

Goldstein, D., Handler, R., and Sirovich, L. (1993). Modeling a no-slip flow boundary with an external force field. J. Comput. Phys., 105(2), 354-366.

Kempe, T., Frölich, J. (2012). An improved immersed boundary method with direct forcing for the simulation of particle laden flows. Journal of Computational Physics, 231(9), 3663-3684.

Kim, J., Kim, D., and Choi, H. (2001). An immersed-boundary finite-volume method for simulations of flow in complex geometries. J. Comput. Phys., 171(1), 132-150.

Lac, C., Chaboureau, J.-P., Masson, V., Pinty, J.-P., Tulet, P., Escobar, J., Leriche, M., and others (2018). Overview of the Meso-NH model version 5.3 and its applications.

Lafore, J. P., Stein, J., Asencio, N., Bougeault, P., Ducrocq, V., Duron, J., Fisher, C., Hèreil, P., Mascart, P., Masson, V., Pinty, J. P., Redelsperger, J.-L., Richard, E., and Vilà-Gueau de Arellano, J. (1998). The Meso-NH Atmospheric Simulation System. Part I: adiabatic formulation and control simulations. Scientific objectives and experimental design. Annales Geophysicae, 16, 90-109.

Lundquist, K. A., Chow, F. K., and Lundquist, J. K. (2010). An immersed boundary method for the Weather Research and Forecasting model. Mon. Wea. Rev., 138(3), 796-817.

Lundquist, K. A., Chow, F. K., and Lundquist, J. K. (2012). An immersed boundary method enabling large-eddy simulations of flow over complex terrain in the WRF model. Mon. Wea. Rev., 140(12), 3936-3955.

Mittal, R., and Iaccarino, G. (2005). Immersed Boundary Methods. Annu. Rev. Fluid Mechs, 37:239-261.

Popinet, S. (2003). Gerris: a tree-based adaptive solver for the incompressible Euler equations in complex geometries. J. Comput. Phys., 190(2), 572-600.

Straka, JM, Wilhelmson, R. B., Wicker, L. J., Anderson, J. R., and Droegemeier, K. K. (1993). Numerical solutions of a non-linear density current: A benchmark solution and comparisons. Int. J. for Num. Methods in Fluids, 17(1), 1-22.

Please also note the supplement to this comment:
https://www.geosci-model-dev-discuss.net/gmd-2018-7/gmd-2018-7-AC1-supplement.zip

––––––––––––––––––––––––––

---

## Author Comment (AC2) · 3 Nov 2018

**Introduction**

We thank the Referee 2 for his/her interest in our work and his/her positive appreciation of the manuscript. We are glad that the Referee 2 gives some suggestions to improve the manuscript. Our response is split in two sections. The first section answers to the similar comments done both by the Referee 1 and by the Referee 2. The second section gives the responses related to the specific comments done by the Referee 2. As the discussion progresses, we invite the Referee 2 to look after the revised version

of the manuscript sent with the present document. A color code applied to the text highlights the modifications: the red color is used when modifications/corrections are done; the green color is used when new insertions are proposed. Note the modification of the title following the GMD requirement.

**1 Common response to Referee 1 and Referee 2**

*The Referees compliment the extensive work but feel that the details of the numerical implementation are not clearly expressed. Moreover, the numerical implementation suffers from a lack of details.*

To propose an immersed boundary method (IBM) in the Meso-NH (MNH) code able to model the ground or topography interaction with an atmospheric flow, as it mentions by the Referees, the number of numerical developments and associated validations has to be high. It induces a long description which could be problematic regarding the format of a manuscript. For this reason, the authors decided to condense the ample information running the risk of loosing a part. The authors agree with the Referees observations. Therefore, in the proposed revised version, an important effort to give additional details is done. About the organization, Referee 1 suggests either to split the paper in two parts and/or to place the current Sections 4, 5 and 6 (which are pointed out in the limit of the GMD scope) in Appendix. Referee 2 requires more details on the cylinder case at moderate Reynolds number (Sect. 6). The authors share the same view and propose to preserve one test case dedicated to the forcing of the pressure solver and to place the others test cases of Sections 4 and 5 in Appendix. Following the Referee 2 and because of the GCT validation, Section 6 is preserved in the core of the paper. The new structure of the paper allows to detail the numerical methods (Sect. 2 and 3) without an increase of the paper volume.

In particular, the Referees make it clear the lack of details on the use of the image points in the Ghost-Cell Technique (GCT) and on the Level-Set Function (LSF). The discussion on GCT in Section 3 is therefore reinforced. Concerning LSF and in the present paper, this function is built for academic bodies and their theoretical solutions are known. The intensive work we had done to implement an accurate LSF was related to the modeling of an interface only known in a discrete way (such as the data of a real urban topography). This work is presented in another paper: Auguste et al.(submitted to Atmospheric Environment). That's said, the LSF presentation in the present paper is reinforced in Section 3.

*The Referees mention that the simulation of the Taylor-Green vortex does not have to appear in the section dedicated to the potential flows.*

The Referees are absolutely right. At the short time scales, the viscous influence vanishes and the Taylor-Green vortex solution is associated to an inviscid flow. The confusion (and mistakes) of the authors to put this test case in a "Potential Flows" category comes from an abusive use of the "Taylor-Green vortex" term. Even if the flow structure solution presents an array of vortex similar to the Taylor-Green ones, we do not have the right to use this term. The studied potential flow (the velocity field derives from $\pi^{-1}cos(\pi lx)sin(\pi my)$) is solution of the Poisson equation (Popinet, 2003). This case testing the pressure solver moves into Appendix of the proposed revised manuscript.

*The Referees mention a lack of details and/or confusion on the molecular diffusion used in the Direct Numerical Simulations.*

The molecular diffusion is taken into account in the cases presenting a low Reynolds number ("low" compared to most of $Re$ of atmospheric applications). This term is associated to the fluid kinematic viscosity $\nu_f$. Therefore, $\nu_f \Delta u$ is explicitly added

in a physical purpose (Navier-Stokes equations resolution, Eq. 3). The numerical implementation is the most simple: in 1D for example, its contribution on the $\frac{\partial u_i^{n+1}}{\partial t}$ is computed for uniform Cartesian grids such as $\nu_f/\Delta^2(u_{i+1}^n - 2u_i^n + u_{i-1}^n)$ where $\Delta$ is the space step. The explicit-in-time resolution induces the respect of the stability condition $\mathcal{O}(\nu_f/\Delta^2)$. Some additional comments about the fluid viscosity are inserted in Section 2. Even if this type of flow is far from the atmospheric applications, this is a robust way to test and validate the implemented GCT. For example, this study makes us confident in the forcing of the Reynolds stresses $\nabla.(\nu_e \nabla \overline{u})$ near an immersed wall ($\nu_e$ the turbulent viscosity). In the same spirit and for future thermodynamic applications, the authors mention that another study was carried out on the parabolic heat equation to confront (and validate) the GCT to a 1D pure diffusion problem $\frac{\partial T_f}{\partial t} = (\lambda_f/\rho C_p)\Delta T_f$.

**2 Specific response to Referee 2**

*Section 3. In Figure 1 (a), what does the triangle mean?*

The triangle symbol indicates an arbitrary type of nodes (P/U/V/W for example). This symbol does not appear in the new Fig.1-a. This symbol is defined in Fig.2-b.

*The authors should give the mathematical expression of the level set function and show how it is used to identify the interface by one example, as did in the cited paper of Tseng and Ferziger (2013).*

To improve the introduction of LSF, a new Fig. 1-a is used. In addition, the definitions of the vector normal to the interface and the curvature are given.

*In the expression $GI = 2phi_G n$, the variable $phi_G$ is not defined.*

As it is suggested, the definitions of $\phi_G$ and $\phi_I$ are added.

*The use of "original" for the proposed method to differ from the classical method would be misleading and inappropriate. The author could use another word like "new", "novel" or "proposed" instead.*

We agree with the Referee and the "original" term is substituted by "new" or "proposed".

*In the flow over a stationary cylinder test, the authors have studied different meshes but not indicated the domain size, while which has much greater impact to the final solution. Comparison could be made to the reference paper "Moving immersed boundary method, International Journal for Numerical Methods in Fluids, 2017". A convergence study could be performed for the convergence rate of velocity, as it never shows in the other parts.*

The domain size is indicated: "The limit of the numerical domain is $10D_{cyl}$ upstream the obstacle for the inlet condition ($U_\infty$, the uniform incoming velocity) and lateral condition (slip condition), $15D_{cyl}$ for the outlet condition allowing the vorticity evacuation." Our domain size choice followed the Auguste (2010) one. That's said, the domain size can have a dramatic influence studying an unbounded Stokes regime for example and we agree with the Referee about the possible domain size influence in the 2D presented case at moderate $Re$. Numerical effects of the inlet/outlet conditions can weakly affect our results at $Re = 40$ as it is demonstrated in Cai et al. (2017); this is one of the reason to use the $\approx$ symbol in Table 1. Even if a convergence study based on a variable such as an axial velocity in the wake of the body does not appear in the proposed revised version, the paper indicates the ability to simulate the physical problem with a good description of the vortex structure in the near-wake of the cylindrical body. To compensate the non-appearance of a convergence study, the proposed revised version of the paper is enriched with a supplementary materials which shows the ability of MNH-IBM

to simulate a physical problem (Straka et al., 1993) governed by thermal effects and viscous effects (the used $\nu_f$ value is $10^4$ higher than the atmosphere one). This study compares with the literature results (body conformal grid method) the velocity and the spread of a density current.

**References**

Auguste, F. (2010). Instabilités de sillage et de trajectoire dans un fluide visqueux. Ph.D. thesis, University of Toulouse.

Auguste, F., Paoli, R., Lac, C., Masson, V., and Cariolle, D. (submitted to Atmospheric Environment). Large eddy simulations devoted to the health impact of pollutant dispersions in cities: the case of the NO2 plume due to the AZF explosion in Toulouse (21/09/01).

Brennen, C. E. (1982). A Review of Added Mass and Fluid Inertial Forces. Naval Civil Engineering Laboratory, Port Hueneme, CA. CR 82.010.

Cai, S.-G., Ouahsine, A., Favier, J., and Hoarau, Y. (2017) Moving immersed boundary method. Int. J. Numer. Methods Fluids, 85(5), 288-323.

Cuxart, J., Bougeault, P., and Redelsperger, J.-L. (2000). A turbulence scheme allowing for mesoscale and large-eddy simulations. Quart. J. Roy. Meteor. Soc., 126(562), 1-30.

Goldstein, D., Handler, R., and Sirovich, L. (1993). Modeling a no-slip flow boundary with an external force field. J. Comput. Phys., 105(2), 354-366.

Kempe, T., Frölich, J. (2012). An improved immersed boundary method with direct forcing for the simulation of particle laden flows. Journal of Computational Physics, 231(9), 3663-3684.

Kim, J., Kim, D., and Choi, H. (2001). An immersed-boundary finite-volume method for simulations of flow in complex geometries. J. Comput. Phys., 171(1), 132-150.

Lac, C., Chaboureau, J.-P., Masson, V., Pinty, J.-P., Tulet, P., Escobar, J., Leriche, M., and others (2018). Overview of the Meso-NH model version 5.3 and its applications.

Lafore, J. P., Stein, J., Asencio, N., Bougeault, P., Ducrocq, V., Duron, J., Fisher, C., Hèreil, P., Mascart, P., Masson, V., Pinty, J. P., Redelsperger, J.-L., Richard, E., and Vilà-Gueau de Arellano, J. (1998). The Meso-NH Atmospheric Simulation System. Part I: adiabatic formulation and control simulations. Scientific objectives and experimental design. Annales Geophysicae, 16, 90-109.

Lundquist, K. A., Chow, F. K., and Lundquist, J. K. (2010). An immersed boundary method for the Weather Research and Forecasting model. Mon. Wea. Rev., 138(3), 796-817.

Lundquist, K. A., Chow, F. K., and Lundquist, J. K. (2012). An immersed boundary method enabling large-eddy simulations of flow over complex terrain in the WRF model. Mon. Wea. Rev., 140(12), 3936-3955.

Mittal, R., and Iaccarino, G. (2005). Immersed Boundary Methods. Annu. Rev. Fluid Mechs, 37:239-261.

Popinet, S. (2003). Gerris: a tree-based adaptive solver for the incompressible Euler equations in complex geometries. J. Comput. Phys., 190(2), 572-600.

Straka, JM, Wilhelmson, R. B., Wicker, L. J., Anderson, J. R., and Droegemeier, K. K. (1993). Numerical solutions of a non-linear density current: A benchmark solution and comparisons. Int. J. for Num. Methods in Fluids, 17(1), 1-22.

Please also note the supplement to this comment:
https://www.geosci-model-dev-discuss.net/gmd-2018-7/gmd-2018-7-AC2-supplement.zip

---

## Author Comment (AC3) · 3 Nov 2018

Dear A. Kerkweg,

To answer to your requests, we have inserted in the title of the revised version of the paper the version number (v5.2) of the Meso-NH code used in the present work. The immersed boundary method described in the paper is implemented in the code such as an additional numerical scheme and therefore we do not think that it is necessary at this step of development to affect a version number for the IBM module. As it mentions in the code availability section, the source files will be accessible on a CERFACS server. Note that the implementation of the IBM in the last version of MNH (v5.4) is currently

in progress.

Best regards, Franck Auguste.
* * *

---

## Author Response (AR2)

**Response to Topical editor. *Implementation of an Immersed Boundary Method in the Meso-NH model: Applications to an idealized urban-like environment.**

F. Auguste[1], G. Réa[1], R. Paoli[1], C. Lac[2], V. Masson[2], and D. Cariolle[1,2]

[1]CECI, CNRS, CERFACS, Toulouse, France
[2]CNRM, CNRS, Météo-France, Toulouse, France

*Correspondence to:* Franck Auguste (franck.auguste@cerfacs.fr)

**Introduction**

We thank the Topical Editor for his interest in our work and we are glad that the Topical Editor gives some suggestions to improve the manuscript, especially on inaccuracies in the use of the English language; we kindly revise the manuscript accordingly. A point-by-point reply to the comments is done in the next section. The reply is supported by the revised and marked-up manuscript versions. The red (resp. blue) color is used in the marked-up manuscript version to highlight the corrections due to the Topical Editor (resp. other native English speaker) comments. Note that, in the previous manuscript version, the introduction was corrected by a native English speaker and the conclusion by a professional translator.

**1  Point-by-point reply**

*p02-l01: space after the comma: ...cal effects ), their p...* The correction is inserted.

*p02-l05 revise the English language: ... fine-scale flow fluctuations can possibly trigger important nonlinear physicochemical processes and should then be captured by the simulations.* We propose: "Furthermore, fine-scale flow fluctuations influence nonlinear physicochemical processes."

*p02-l15: is Moriwaki and Kanda, 2004) the right reference for COSMO project? This paper refers to fluxes of heat and CO2 in a suburban area of Tokyo.* The research groups of the COSMO scale model project had conducted continuous tower measurements called Kugahara Project in a densely built up residential area in Japan. This project had been enhanced by the Moriwaki and Kanda (2004). We propose : "COSMO (Comprehensive Outdoor Scale Model Experiment for Urban Climate) and Kugahara projects (Moriwaki and Kanda, 2004)"

*p02-l21 Please cut the phrase: "Covering all possible cases is obviously impossible but from a fluid mechanics standpoint one can invoke the persistence of flow behaviors whatever the 25 scales allowing for example to observe von Kármán streets in the wake of a centimeter-scale cylinder as well as in the cloud layout behind an island." is too long.* We propose: Covering all possible cases is obviously impossible. From a fluid mechanics standpoint one can invoke the persistence of flow behaviors whatever the scales. For example, the von Kármán streets are observed in the wake of a centimeter-scale cylinder as well as in the cloud layout behind an island.

*p02-l26 urban cities sounds repetitive, urban areas or cities.* The "urban areas" formulation is inserted.

*p03-l04 clarify "This approach can suffer from the time step restriction."* The non-moving boundary approach is obtained by Goldstein(1994) with a moving boundary approach. A strong wall stiffness is modeled inducing a time step limitation. We propose to add: "(spring and damper model with large stiffness value)"

*p03-l22 future mesoscale application, > future mesoscale applications,* The correction is inserted.

*p03-l23 unresolved obstacles such as vegetations > unresolved obstacles such as vegetation.* The correction is inserted.

*p04-l01 resolution is ranging from the > resolution ranges from the* The correction is inserted.

*p04-l14 Please do not start a phrase with a symbol (rho). I suggest to introduce them later, after used the first time.* The density and the potential temperature of the reference state are introduced later.

*p04-l16 clarify the meaning of "law"* We propose to add: "(partial differential equations, continuum mechanics)"

*p04-l18 There is no resp. for the unresolved part. Please rewrite.* You're right. We propose: "into a resolved part and unresolved one".

*p04-l26 transport is not to be used in plural in this phrase. Use "the transport for each...* The correction is inserted.

*p04-l29 same: scalar transport* The correction is inserted.

*p05-l06: clarify the meaning of n in the Courant number* We propose to add: "($n$, the time step index)".

*p05-l13: meaning of n+1 in the continuity equation* The previous insertion defines $n$.

*p05-l13: It seems you n define as the time step index in P7* Yes.

*p05-l21: The mathematical operator to inverse > The mathematical operator to invert?* You're right. The correction is inserted.

*p06-l15: only the turbulent fric- tion is claimed. > only the turbulent fric- tion is used.?* The correction is inserted.

*p06-l21: terms is neglected? > "terms are neglected" or "term is neglected"* The correction "term is neglected" is inserted.

*p06-l24: "acting; a" better use and intstead of semicolon.* The correction "acting and a" is inserted.

*p06-l27: "LSF LevelSet Function: in general, please use parenthesis when introducing acronyms* Parenthesis are inserted.

*p07-l06: "An intensive study had been done to verify the ability of the LSF spatial derivatives to recover the vector normal to the interface and the local curvature (Auguste et al., Submitted)." Check the English language, and cite only the finally published version of the reference.* The reference is not currently published (under review). Therefore, the reference disappears. We rephrase the sentence: "An intensive study had been done to estimate the well-modelling of the vector normal to the interface and the local curvature using LSF."

*p08-l06: had also been tested please check the tense use.* "tested" is replaced by "implemented".

*p08-Figure 2 caption: please use parenthesis when defining a symbol e.g. ghost (G), etc.* The corrections are inserted.

*p08-l11: please revise equations within the phrase. Do not start a phrase with an equation. Leave spaces after points.* The phrase ": $\phi(i,j).\phi(i,[j-k_l:j+k_l]) > 0$ and $\phi(i,j).\phi([i-k_l:i+k_l],j) > 0$ ($k_l$ an integer, 2D case). $k_l = 1$ (resp. $k_l = 2$) allows..." is re-written: ". For a 2D case, the sign of $\phi(i,j).\phi(i,[j-k_l:j+k_l])$ and $\phi(i,j).\phi([i-k_l:i+k_l],j)$ is estimated. The integer value $k_l = 1$ (resp. $k_l = 2$) allows..."

*p08-l12: What to you mean with: Such mage-ghost distance...* "mage-ghost distance" is replaced by "ghost layer".

*p08-l14: Please clarify the meaning of the condition.* We propose to insert: "The stencil of the numerical schemes modelling the interface defines the $k_l$ value. Therefore and to limit this overhead,..."

*p09-l02: Please do not start a phrase with a symbol.* "$| IB |$ ... " is rephrased by: "The $| IB |$ distance.."

*p09-l04: The physical information at I is directly related to the interfaces one which can itself be dependent on the fluid information Please revise English language, especially the use of the possessive. p09-l05: Revise also (as its done in the wall models used in LES). Replace its* We propose to insert "leading to a not well pose condition"

*p09-Please revise the whole paragraph: it is important because it explains the need of and justifies the new development.*

We propose to insert more details in this paragraph:" To overcome this problem, another way is to define image points (noted $I_1$ and $I_2$ in Fig. **??**-a, merely renamed images) having a distance to the interface $\Delta$-dependent and not $\phi_G$-dependent: $\mathbf{GI_l} = l\Delta + \phi_G\mathbf{n}$ with $l = (1; 2)$. Figure **??**-a shows the images for one ghost. The new approach enforces a large enough value of the $|\mathbf{I_1B}|$ distance. Figure **??**-b (resp. c) illustrates the classical (resp. original) GCT for several ghosts. Figure **??**-b shows some mirror points associated with ghost of the first layer in the vicinity of the interface. Whatever the ghost location, Figure **??**-c shows the new approach ensuring the image points to be embedded in the fluid region. The definition of several images per ghost allows to build a profile of the $\overline{\psi}$ fluid information normal to the interface. $\overline{\psi}(I)$ is therefore recovered by a quadratic reconstruction $f$ using the $(B, I_1, I_2)$ points. "

"The accuracy of an interpolation is depending on the $\overline{\psi}$-profile. For example, a logarithmic evolution of the tangent velocity is expected in LES. Otherwise when the viscous layer is modelled, a linear evolution is expected. To compare the ability of each quadratic interpolation to approach a wide variety of profiles, the recover of power laws such as $\psi = \phi^{3/2}$ (Figure **??**-a) and $\psi = \phi^{1/4}$ (Fig. **??**-b) is studied. As it illustrates, $PLI^a$ fits better the two analytical solutions and is therefore adopted."

*p11-l05: This yields induce: revise* The phrase is replaced by: "The cotangent direction is defined such as:".

*p11-eq 20: use text font in equation mode.* The text font is used.

*p14-l11: to an unwell posed system > use not well pose instead.* The correction is inserted.

*p17-l05: It will raise in the future issues on the validity of the log-law near a singularity (sharp edges or corners). please revise.* We propose: "Note that the use of a log-law model near a singularity such as sharp edges and corners could be called into question."

*p17-l24: The RICH Richardson and the RESI Residual Conjugate Gradient  Please use parenthesis when defining symbols.* Parenthesis are inserted.

*p17-l27: Please do not start a phrase with an equation or symbol.* "The slope coefficient" is inserted.

*p17-Fig 10 please provide a higher quality image.* The old resolution of the picture was 1000*490 pixels (png format). The new resolution is 1420*700 pixels (png format).

*p22-l02: had provided check tense.* We propose to replace this by "had collected".

*p22-l15: Please do not start a phrase with a symbol. E.g. The stream-wise (resp. span-wise) direction is x* The proposition is inserted.

*p23-l06: Please revise English language: The literatures LES give the* We propose to replace this by "The LES reference results".

*p23-l03: Please revise English language: and as the literature mentions, it presents* We propose to replace this by "and as the LES reference results mention, it presents".

*p26-l14: The towers location is > The locations of the towers are* The correction is inserted.

*p27-l11: What do you mean by comforted by?* We propose to replace this by "close to".

*p28-l03: (numerical confinement, under investigations). > (numerical confinement, under investigation).* The correction is inserted.

*p28-l15: two times greater > two times larger* The correction is inserted.

*p28-l16: Do not use the dot as multiplication symbol in the units.* The correction is inserted.

*p29-l08: of ongoing works in > of ongoing work in* The correction is inserted.

*p30-l06 of the GCT Ghost-Cell etc. Please use parenthesis when defining an acronym.* The parenthesis are inserted.

*p30-l19-22: long Phrase, consider rearranging in more sub-phrases.* We propose: "These tests validate the proposed GCT 'three images/ghost points' using an inverse distance weighting (resp. trilinear) interpolation near (resp. far from) the interface. The modelling of the advection term near the fluid-solid interface is ensured by a 2nd-order centered scheme associated with an artificial viscosity. The artificial viscosity is calibrated after comparisons with a 3rd-order WENO scheme."

*p30-l28: Future works. > Future work.* The correction is inserted.

35    *p30-l31: high-curvatures bodies > high-curvature bodies* The correction is inserted.

*p30- non-flat grounds. > non-flat ground.* The correction is inserted.

*p30-l34: with works dedicated with work dedicated* The correction is inserted.

*p31-l03: compliances with > compliance with* The correction is inserted.

*p31-l03: Such compliances > Such compliance* The correction is inserted.

*p31-l03: Table ?? Please fix missing ref* A bracket was missing. Thanks.

**References**

[revised manuscript text omitted]

---

## Author Response (AR3)

**Response to Topical editor. *Implementation of an Immersed Boundary Method in the Meso-NH model: Applications to an idealized urban-like environment.**

F. Auguste[1], G. Réa[1], R. Paoli[1], C. Lac[2], V. Masson[2], and D. Cariolle[1,2]

[1]CECI, CNRS, CERFACS, Toulouse, France
[2]CNRM, CNRS, Météo-France, Toulouse, France

*Correspondence to:* Franck Auguste (franck.auguste@cerfacs.fr)

**Introduction**

Once more we thank the Topical Editor for his interest giving some suggestions to improve the manuscript. We kindly revise the manuscript accordingly. A point-by-point reply to the comments is done in the next section. The reply is supported by the revised and marked-up manuscript versions. The red color is used in the marked-up manuscript version to highlight the corrections. The last section gives some comments on the code avalaibility.

**1   Point-by-point reply**

*p03-l06: (spring and damper model with large stiffness value). please further clarify the explanation of the time step restriction.* The stability of the interface equation is affected by the stiffness $K$ resolved by an explicit method. The time step restriction is associated to the natural frequency $F$ of the interface $F \sim \mathcal{O}(K^a)$ where $a$ is a positive real; a high frequency $F$ induces a small time step used in the system. We propose: '(spring and damper model for an interface with large natural frequency)'

*p03-l10: Many types of discrete forcing exist and a non exhaustive list can be: direct forcing in the fluid region near the interface as in Mohd-Yusof (1997), immersed interface method (Leveque and Li, 1994), Cartesian grid method (Clarke et al., 1986). Many types of discrete forcing exist, e.g. direct forcing in the fluid region near the interface (Mohd-Yusof , 1997), immersed interface method (Leveque and Li, 1994), Cartesian grid method (Clarke et al., 1986), etc. (the list is not exhaustive).* The correction is inserted.

*p03-l10: An argument in favor of GCT is that it does not introduce source terms in the conservation equations so that boundary conditions are imposed at the interface and/or in the solid region, the only corrections to the physical model come from subgrid turbulent parameterizations, and boundary condition is imposed at the interface and/or in the solid region. This phrase is still cryptic. Cut and clarify.* We propose: The GCT does not introduce source terms in the conservation equations so

that boundary conditions are imposed at the interface and/or in the solid region. The only corrections to the physical model in the fluid region come from subgrid turbulent parameterizations.

*p04-l30: corresponds* The correction is inserted.

*p05-l07: by either* The correction is inserted.

*p05-l25: to inverse->invert* The correction is inserted.

*p06-l22: I do not understand the meaning of the equation in l 22 (for $u_i w$). Is the equation well formatted? What is the meaning of the vertical bars?* The equation is well formatted but the notation is not clear. $u$ and $v$ are currently defined such as the horizontal components of the velocity field; $w$ the vertical one. We propose to conserve the notation of Eq. 6 replacing $u'w'$ and $v'w'$ by $u'_i u'_j$. $\mid X \mid$ correspond to the absolute value of $X$. The length of the vertical bar is modified.

*p08-l02: Never start a phrase with a symbol (The value .. is ..)* The correction is inserted.

*p08-l07: Again start a phrase with a symbol* The correction is inserted.

*p14-l13: to inverse->invert* The correction is inserted.

*p15-l03: not well pose system > is not well posed.?* Yes, the correction is inserted.

*Sect. 7: Remember that the article cannot be published if the authors do not make their code available. The available options are to submit the code with the manuscript and to provide a doi.The executive editors clearly state: "The paper must be rejected if the authors refuse to comply with requests to make the code accessible within the requirements of GMD." Add:*
*- Code (including proposed immersed wall method of section 3.3)*
*- input for validation with potential and viscous flows (4.1, 4.2)*
*- input for turbulent flow simulations (5.1, 5.2)*
We propose: "The source files dedicated to IBM and the input files for the simulations of Sections 4 and 5 are accessible on a CERFACS server: https://cerfacs.fr/MNHIBM/Auguste-GMD-2019."

The *Auguste-GMD-2019* directory contains the modifications of the source code. The sub-directory *INPUTS* contains the namelists for the simulations presented in the core of the paper. The classification of these input files (preparation and execution) respect the one of the paper sections.

[revised manuscript text omitted]

---

## Author Response (AR4)

**Response to Topical editor. *Implementation of an Immersed Boundary Method in the Meso-NH model: Applications to an idealized urban-like environment.**

F. Auguste[1], G. Réa[1], R. Paoli[1], C. Lac[2], V. Masson[2], and D. Cariolle[1,2]

[1]CECI, CNRS, CERFACS, Toulouse, France
[2]CNRM, CNRS, Météo-France, Toulouse, France

*Correspondence to:* Franck Auguste (franck.auguste@cerfacs.fr)

**Introduction**

Once more we thank the Topical Editor for his interest giving some suggestions to improve the manuscript. We kindly revise the manuscript accordingly. A point-by-point reply to the comments is done in the next section. The reply is supported by the revised and marked-up manuscript versions. The red color is used in the marked-up manuscript version to highlight the
5   corrections.

**1   Point-by-point reply**

*Thank you for preparing the code associated with the manuscript and uploading it to the webpage:*
   *https://cerfacs.fr/MNHIBM/Auguste-GMD-2019. Please notice that making code available in external webpages does no longer comply with GMD standards. Please either upload the code to a repository that provides DOIs (such as Zenodo) or*
10   *alternatively, provide a tar file with the contents of https://cerfacs.fr/MNHIBM/Auguste-GMD-2019 as supplementary material of this article if the size of the tar ball is reasonable.*

   The source files dedicated to IBM and the input files for the simulations of Sections 4 and 5 are uploaded on the webpage https://cerfacs.fr/MNHIBM/Auguste-GMD-2019. The *Auguste-GMD-2019* directory contains the modifications of the Meso-
15   NH source code and the IBM implementation in this code. The sub-directory *INPUTS* contains the namelists for the simulations presented in the core of the paper. The classification of these input files (preparation and execution) respect the one of the paper sections.
   In addition, the tar ball https://cerfacs.fr/MNHIBM/Auguste-GMD-2019.tar has been created. This file can be directly download on the webpage. The size of the tar ball (1Mo) is reasonable and therefore can be a supplementary material of this article.

[revised manuscript text omitted]